# PIVOT-R: Primitive-Driven Waypoint-Aware World Model for Robotic Manipulation

Kaidong Zhang[1]*    Pengzhen Ren[2]*    Bingqian Lin[1]    Junfan Lin[2]

Shikui Ma[3]    Hang Xu[4]    Xiaodan Liang[1,2]†

[1]Sun Yat-sen University    [2]Peng Cheng Laboratory    [3]Dataa Robotics    [4]Huawei Noah's Ark Lab

https://abliao.github.io/PIVOT-R

## Abstract

Language-guided robotic manipulation is a challenging task that requires an embodied agent to follow abstract user instructions to accomplish various complex manipulation tasks. Previous work generally maps instructions and visual perceptions directly to low-level executable actions, neglecting the modeling of critical waypoints (*e.g.*, key states of "close to/grab/move up" in action trajectories) in manipulation tasks. Trivially fitting the data without revealing the relation between instruction and low-level executable actions, these models are prone to memorizing the surficial pattern of the data instead of acquiring the transferable knowledge, and thus are fragile to dynamic environment changes. To address this issue, we propose a **PrI**mitive-dri**V**en wayp**O**in**T**-aware world model for **R**obotic manipulation (PIVOT-R) that focuses solely on the prediction of task-relevant waypoints. Specifically, PIVOT-R consists of a Waypoint-aware World Model (WAWM) and a lightweight action prediction module. The former performs primitive action parsing and primitive-driven waypoint prediction, while the latter focuses on decoding low-level actions. Additionally, we also design an asynchronous hierarchical executor (AHE) for PIVOT-R, which can use different execution frequencies for different modules of the model, thereby helping the model reduce computational redundancy and improve model execution efficiency. Our PIVOT-R outperforms state-of-the-art (SoTA) open-source models on the SeaWave benchmark, achieving an average relative improvement of 19.45% across four levels of instruction tasks. Moreover, compared to the synchronously executed PIVOT-R, the execution efficiency of PIVOT-R with AHE is increased by 28-fold, with only a 2.9% drop in performance. These results provide compelling evidence that our PIVOT-R can significantly improve both the performance and efficiency of robotic manipulation.

## 1 Introduction

Language-guided robotic manipulation [22, 33, 61, 50, 12, 38] is a key research problem of Embodied AI. This field aims to enable agents to follow abstract language instructions for performing various manipulation tasks. To complete the tasks, the agent needs to transform high-level language instructions into low-level actions as well as capturing environmental dynamics for precise manipulation decision-making.

Witnessed the immense success of vision-language foundation models (VLMs) [2, 40, 37], many works have explored the utilization of VLMs for facilitating language-guided robotic manipulation in

---

*Equal contribution
†Corresponding authors

38th Conference on Neural Information Processing Systems (NeurIPS 2024).

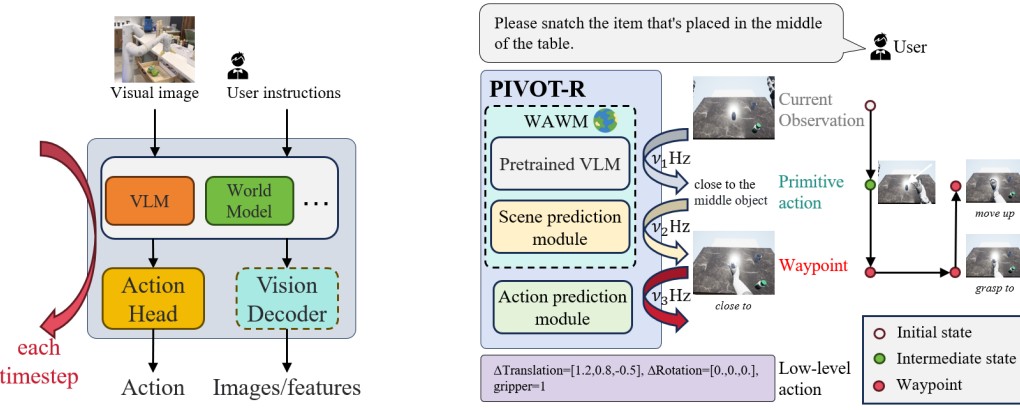

(a) Sequentially executed robot manipulation model      (b) PIVOT-R

Figure 1: Comparison of PIVOT-R and other models. (a) Sequentially executed robot manipulation model. They sequentially execute each module in the model at each timestep to perform manipulation reasoning (*e.g.*, RT-2 [64], RT-X [49], RT-H [5], VILA [20], Octo [36], *etc*.) or world modeling (*e.g.*, Surfer [42], Daydreamer [56], 3D-VLA [60], *etc*.) This easily leads to model redundancy and weak key manipulation node prediction capabilities. (b) PIVOT-R is a primitive-driven waypoint-aware world model with asynchronous hierarchical executors. It only focuses on the prediction of waypoints related to the manipulation task, and it is easier to predict key nodes in the manipulation task than other methods. In addition, PIVOT-R sets different execution frequencies for different modules to have higher execution efficiency and lower redundancy.

recent years [48, 64, 49, 27, 21, 20]. For example, RT-2 [64], RT-X [49], RT-H [5], and RoboFlamingo [27] employ the VLM as the backbone and introduce large-scale vision language data for manipulation training, which significantly improve the generalization. VILA [20] resorts to GPT-4 [37] to generate sequential actionable steps for improving long-horizon planning. In addition, 3D-VLA [60] and Daydreamer [56] have also tried to introduce world models into robot manipulation to help the models free themselves from a large amount of trial and error and improve learning efficiency. Despite extensive efforts made by researchers, two key challenges remain: *(i)* Weak key waypoint prediction and world modeling capabilities; *(ii)* High computational redundancy and inefficient execution.

For the first challenge, such as "moving a cup", humans intuitively apply their internal world models to seamlessly analyze and predict task-related key action flows: "getting close to the cup → grab the cup → move the cup → put down the cup". Similar to the approaches in navigation tasks, we define these key action frames as waypoints for manipulation tasks. Figure 1 (b) right shows a robot manipulation task with three waypoints. How to enable robots to acquire this ability is very critical. To this end, RT-H [5] uses VLM to perform natural language parsing of key action nodes and uses language to guide robot manipulation. However, it does not perform world modeling on visual scene information. Therefore, some work [56, 34, 60, 42] have attempted to summarize general dynamic knowledge about the environment and predict future outcomes by introducing world models, to generate more executable long-term plans and accurate manipulation action decisions. However, they tend to model the world at each timestep of robot manipulation, leading to the neglect of waypoints which have a more direct impact on manipulation success. To make matters worse, in the long-term lack of key waypoint guidance, the randomness of each action prediction may be continuously amplified due to the existence of low-level action similarities under local spatiotemporal conditions.

For the second challenge, as shown in Figure 1 (a), previous methods [64, 5, 49, 42, 56] tend to treat different modules in the model equally and execute all modules sequentially, which is not necessary and inevitably introduces redundancy of computation and causes a great cost of resources. To this end, MResT [43] proposes a multi-resolution transformer that uses different execution frequencies for different spatial and temporal resolutions to control coarse, precise, and dynamic tasks in real-time, thereby effectively reducing unnecessary computational redundancy and improving the real-time performance of robot manipulation. However, it lacks focus on world modeling capabilities and cannot predict critical nodes of manipulation tasks as accurately as humans.

Based on the above observations, as shown in Figure 1 (b), in this paper, we propose PIVOT-R, a primitive-driven waypoint-aware world model with an asynchronous hierarchical executor for robot manipulation. PIVOT-R mainly consists of a waypoint-aware world model (WAWM) and an action

prediction module. Specifically, in WAWM, we first use the pre-trained VLM for primitive action parsing and use it as a primitive prompt for the scene prediction module to help the model perform modeling of the robot manipulation waypoint scene. Then, we use waypoints as cues for low-level action prediction. Thanks to WAWM's modeling of key waypoint information, PIVOT-R achieves an average relative performance improvement of 19.45% compared to the state-of-the-art (SoTA) open-source manipulation model on SeaWave's [42] 4-level instruction tasks. In addition, to reduce model redundancy, we also design an asynchronous hierarchical executor (AHE) for PIVOT-R, which sets a slow-to-fast execution frequency scheduler for the three modules of primitive action parsing, scene prediction, and action prediction in the model to help PIVOT-R improves execution efficiency. With the help of AHE, the execution efficiency of PIVOT-R integrated with VLM has not dropped significantly. Compared with synchronously executed PIVOT-R (all modules use the same execution frequency), the execution efficiency of PIVOT-R with AHE is increased by 28 times, while the performance only drops by 2.9%. Our contributions can be summarized as follows:

- We show that modeling of waypoints prevents critical robot dynamics from being submerged in trivial robot manipulations, allowing models to benefit from enhanced dynamic environment modeling.

- The proposed AHE significantly improves the execution efficiency of the model by setting different frequencies for different modules.

- Extensive experimental results demonstrate that our PIVOT-R achieves significantly better performance than the SoTA baseline, such as Gato [41] and RT-1 [7], in all settings.

## 2 Related Work

**Language-Guided Robotic Manipulation.** Robotic Manipulation is a long-standing research field in Embodied Artificial Intelligence. Benefiting from the flexibility and practicality of facilitating human-robot interaction, language-guided robotic manipulation has gained extensive research attention in recent years. Many benchmarks have been built to encourage the research of language-guided robotic manipulation, such as RLBench [22], CALVIN [33], VLMBench [61], *etc*. Early methods improve the manipulation performance by introducing powerful representations [9, 59], elaborated network architectures [15, 13], or effective training mechanisms [32, 44]. With the rapid development of VLMs [2, 40, 37], recent works have attempted to introduce VLMs to improve the manipulation accuracy and generalization to unseen scenarios/objects in a trainable [48, 64, 49, 27, 26] or offline [21, 20, 35] manner. However, most previous approaches tend to learn a direct mapping from multi-modal inputs to low-level actions, ignoring the explicit modeling of environmental dynamics. As a result, they may fail to make executable actions or plans and not generalize well to complex environments. We have also noticed previous work on waypoints and primitive actions, but they often used a limited number of actions. For instance, CLIPort [45], Transporter [57], GMRT [47], and VPG [58] are restricted to simple actions like pick/place/push, limiting their use in complex tasks. Some language-guided models [10, 16, 30] define a few primitive actions ($\leq 5$) and add prompts to aid decision-making. PerAct [46], RVT [14] use robot states as waypoints to skip trivial action predictions. SUSIE [6] and UniPi[11] predict sub-goals through video predictors, but there is an inconsistency between the predicted video and actions. In this work, we propose a waypoint-aware world model to track key dynamics that happen during the manipulation. Our model fulfills asynchronous world modeling and action prediction, which significantly promotes both manipulation accuracy and efficiency. PIVOT-R supports 10 primitive actions and is extensible, making it effective in complex tasks.

**World Models.** World models aim to generate a predictive model of its surroundings, accounting for uncertainties and dynamic changes. They have been widely studied in video generation [4, 53, 8], navigation [51, 24, 39], and autonomous driving [52, 62, 54] areas. For example, Genie [8] introduces a spatiotemporal video tokenizer and a dynamics model to autoregressively predict the next video frame. DriveDreamer [52] builds a world model deriving from real-world driving scenarios for enabling reasonable driving policy generation. With the great potential for acquiring insights into real-world motion and physics rules, some works have also introduced world models for robotic manipulation tasks [56, 34, 60]. Daydreamer [56] applies the Dreamer [17] algorithm to train real-world robots by online reinforcement learning. SWIM [34] collects human videos for training a world model and fine-tuning it on a small amount of robot data. Nevertheless, they usually perform world modeling and decision-making alternatively, bringing great difficulty for training and is also

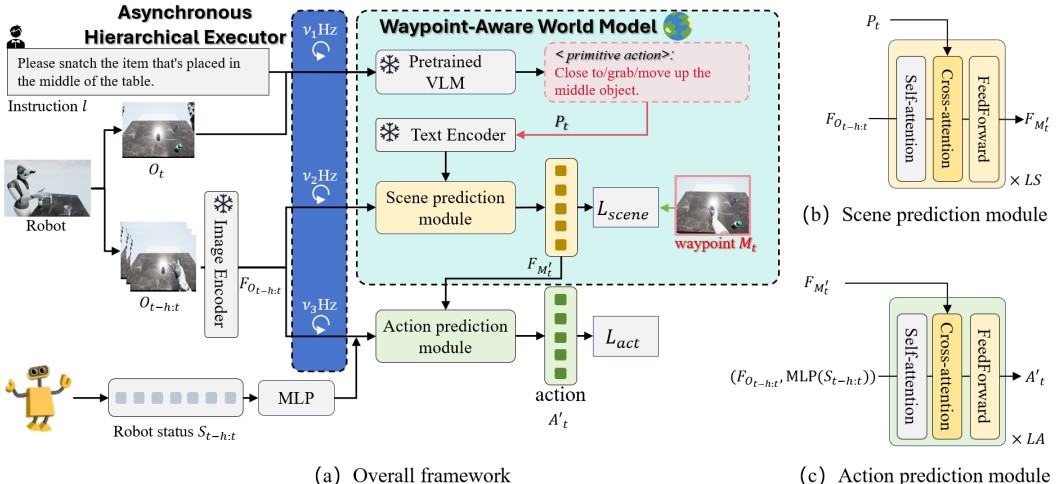

(a) Overall framework     (b) Scene prediction module     (c) Action prediction module

Figure 2: PIVOT-R overview. It mainly consists of a waypoint-aware world model (WAWM) and an action prediction module, where two modules cooperate with each other through an asynchronous hierarchical executor (AHE). In WAWM, we first use pre-trained VLM to perform low-frequency primitive action parsing on user instructions and provide waypoint indications for the scene prediction module. Then, the scene prediction module learns to model the world knowledge based on waypoints and manipulation trajectories. Finally, we use a lightweight action prediction module to perform high-frequency action prediction and execution.

inefficient. In contrast, our proposed WAWM only predicts task-relevant waypoints, empowering realistic and efficient world modeling for improving manipulation performance.

**Vision-Language Foundation models.** Vision-Language Foundation models (VLMs) [2, 40, 37] have witnessed striking advancements in recent years. The ability to understand multimodal inputs and rich real-world knowledge storage of VLMs makes them highly adaptable for a wide range of downstream applications such as image captioning [25, 63] and visual question answering [29, 25]. Many works have also explored the utilization of VLMs in robotic manipulation tasks recently [48, 64, 49, 27, 21, 20]. MOO [48] leverages a pre-trained vision-language model to improve zero-shot open-world object manipulation. RoboFlamingo [27] builds a vision-language manipulation framework upon the open-source VLM OpenFlamingo [2]. VILA [20] and CoPa [21] unleash the commonsense knowledge of GPT-4 for generating accurate and reasonable manipulation action decisions. In this work, we develop an elegant combination of VLMs and world models for tackling the challenging language-guided robotic manipulation task, where we query the VLM, the world model, and the action execution model in an asynchronous way.

## 3 Architecture

Our goal is to build a robot manipulation model that can respond accurately and timely to user instructions in various zero-shot complex and variable environments. To this end, as shown in Figure 2, we introduce a primitive-driven waypoint-aware world model for robot manipulation. Next, we discuss the structural details of each module of PIVOT-R in detail.

### 3.1 Problem Formulation

As shown in Figure 2 (a), we formulate the proposed PIVOT-R as learning a trainable robot manipulation model $\pi$, which maps the user's language instruction $l$ and a series of observation images $O_{t-h:t}$ and robot state $S_{t-h:t}$ from the time step $t - h$ to the current time step $t$ to action $A_t$. $h$ represents the length of the historical frames, here it is set to 3. In addition, we also introduced a scene prediction module for WAWM to help the model model world knowledge. The overall formulation of PIVOT-R is as follows:

$$\pi(\text{VLM}(l, O_t), O_{t-h:t}, S_{t-h:t}) \rightarrow M'_t, A'_t, \tag{1}$$

where $M'_t$ and $A'_t$ are the waypoints and actions of the robot manipulation predicted by the model at timestep $t$, respectively. In particular, we use the pre-trained VLM to parse the primitive actions $P$

the current robot should take from the user instruction $l$ based on the robot's observation image $O_t$. Then, we use $P$ as a waypoint indication for robot manipulation at time step $t$, helping the robot to build prediction and modeling capabilities for future scene information and world knowledge. For each action trajectory $Tra$, it consists of a language instruction $l$ and a series of observation images $O$, robot status $S$, actions $A$, and waypoints $M$:

$$Tra = \{l, [O_1, S_1, A_1, M_1], ..., [O_T, S_T, A_T, M_T]\}, \tag{2}$$

where $T$ is the timestep length of the robot's manipulation trajectory. Note that because we use AHE, the primitive actions $P$ input to the scene prediction module at different time steps $t$ may be the same. The model can avoid redundancy caused by the alternating use of VLM and world models through low-frequency primitive action parsing, thereby improving training and inference efficiency. We adopt similar settings on the action prediction module to further improve the efficiency of the model.

## 3.2 Inputs and Outputs

We provide a detailed description of the inputs and outputs of PIVOT-R in Figure 2 (a) as follows:

- **Language input.** The user's language instruction $l$ is first combined with the prompt and used as the input of the pre-trained VLM to parse the primitive action represented by the short text. The details of the prompt are shown in Appendix F.1. Specifically, in the example of the language instruction "Give me a container of drinking water", the primitive action at this time may be "approach/grab/put down the container". Then, the parsed primitive action and original instruction $l$ are encoded by a text encoder as a text sequence $P$. Following [45, 46, 42], we employ pre-trained CLIP [40] as the language encoder $E_{\text{text}}$.
- **Visual input.** For visual observation of RGB image $O$, we use a pre-trained CLIP [40] visual encoder $E_{\text{image}}$ for encoding.
- **Robot state input.** The robot state includes 6 dimensions of robot arm movement $S = (x, y, z, \text{roll}, \text{pitch}, \text{yaw})$. We use linear layers to encode them.
- **Outputs.** The output of PIVOT-R is the feature $F_{M_t'} \in \mathbb{R}^{b \times n \times d}$ of the task-related waypoint image predicted by the scene prediction module and the robot action $A_t'$ predicted by the action prediction module. Where $b$, $n = 49$, and $d = 512$ represent the batch size, number of tokens, and dimension of the feature $F_{M_t'}$, respectively. The action $A$ contains the delta state $S$ of the robot's end-effector and the binary state $G \in \{0, 1\}$ of the gripper, *i.e.*, $A = (S, G) \in \mathbb{R}^{1 \times 7}$.

## 3.3 Network

Overall, PIVOT-R consists of a powerful waypoint-aware world model and a lightweight action prediction module, whose detailed information is described as follows:

- **Waypoint-Aware World Model (WAWM).** By introducing waypoints as a data structural chunking mechanism, similar to tokenization in NLP, we segment dense and irregular robot trajectories into meaningful sections, reducing the prediction burden. This hierarchical approach decouples language-action interdependencies and leverages cross-trajectory waypoint transition knowledge, improving action prediction accuracy. As shown in Figure 2, WAWM mainly includes a powerful VLM and a scene prediction module $\Phi_{sp}$. Given a user instruction $l$, the VLM parses $l$ to provide task-related waypoint prompts, which are used for guiding the scene prediction module $\Phi_{sp}$ to conduct critical waypoint prediction.

  Specifically, at each timestep $t$, we combine the prompts with the user instructions $l$ and the robot observation images $O_t$ as the input of the pre-trained VLM to perform primitive action parsing related to the manipulation task. Then, the parsed primitive actions and the original user instructions $l$ are combined as waypoint indicators $P_t$ for the scene prediction module. The above process can be expressed as:

$$P_t = (l, \text{VLM}(\text{Prompt}(l), O_t)). \tag{3}$$

  For the scene prediction module $\Phi_{sp}$, we use the waypoint waypoints $P_t$ related to the robot manipulation task as a prompt and the historical observation image $O_{t-h:t}$ of the robot as input to predict the waypoints feature $F_{M_t'}$ of the robot manipulation, that is, we have:

$$F_{M_t'} = \Phi_{sp}(E_{\text{text}}(P_t), E_{\text{image}}(O_{t-h:t})). \tag{4}$$

Table 1: Success rate and speed comparison of different methods in four levels of tasks (%).

| Model | Level 1 | Level 2 | Level 3 | Level 4 | Mean | Time(ms) |
|-------|---------|---------|---------|---------|------|----------|
| Gato [41] | 34.74 | 30.53 | 23.16 | 20.00 | 27.11 | 139 |
| BC-Z [23] | 41.05 | 32.63 | 23.16 | 25.26 | 30.53 | 12 |
| Octo [36] | 69.79 | 48.48 | 34.69 | 33.58 | 46.64 | 18 |
| SUSIE [6] | 78.89 | 48.48 | 32.50 | 29.17 | 47.26 | 434 |
| RT-1 [7] | 67.38 | 49.47 | 38.95 | 34.74 | 47.64 | 21 |
| GR-1 [55] | 77.08 | 55.56 | 37.31 | 34.33 | 51.07 | 35 |
| Surfer [42] | 74.74 | 61.05 | 45.26 | 37.89 | 54.74 | 24 |
| PIVOT-R | **88.06** (13.32 ↑) | **77.55** (16.50 ↑) | **73.33** (28.07 ↑) | **57.82** (19.93 ↑) | **74.19** (19.45 ↑) | 27 |

The model details of the scene prediction module are shown in Figure 2 (b). It is stacked by $LS = 12$ transformer layers. Each transformer layer consists of a self-attention layer, a cross-attention layer, and a feed-forward layer.

- **Action Prediction Module.** For the action prediction module $\Phi_{ap}$, we use the robot manipulation waypoint state features $F_{M'_t}$ predicted by the scene prediction module as prompts, and take the robot's historical observation images $O_{t-h:t}$ and robot status $S_{t-h:t}$ as input to predict the action $A'_t$ that the robot should take at time $t$. Therefore, the prediction process of action $A'_t$ can be expressed as:

$$A'_t = \Phi_{ap}(F_{M'_t}, E_{\text{image}}(O_{t-h:t}), \text{MLP}(S_{t-h:t})). \tag{5}$$

The details of the action prediction module are shown in Figure 2 (c), which has the same structure as the scene prediction module consisting of a stack of $LA = 3$ transformer layers.

## 3.4 Asynchronous Hierarchical Executor

In addition, in order to improve the execution efficiency of PIVOT-R, we adopt an asynchronous hierarchical execution mode to execute primitive action parsing, scene prediction, and action prediction respectively. Specifically, as shown in Figure 2 (a), we use different execution frequencies for these three parts according to needs. For primitive action parsing, it requires a lot of computation using VLM so we use a lower execution frequency $v_1$. For the lightweight action prediction module, we adopt a higher execution frequency $v_3$. These three execution frequencies conform to the following relationship: $v_1 < v_2 < v_3$, where $v_2$ is the execution frequency of the scene prediction module. Specifically, at timestep $t$, if a module has not finished processing the new request, it will return the previous result first.

## 3.5 Loss

The training loss of PIVOT-R mainly includes scene prediction loss $\mathcal{L}_{\text{scene}}$ and action prediction loss $\mathcal{L}_{\text{act}}$. Specifically, for scene prediction loss $\mathcal{L}_{\text{scene}}$, following I-JEPA [1], we calculate the average $L_2$ distance of features between the predicted waypoint state $M'$ and the ground truth $M$, where $M$ is encoded using a pre-trained CLIP image encoder $E_{\text{image}}$. For action prediction loss $\mathcal{L}_{\text{act}}$, following RT-1 [7], we use Cross Entropy Loss to calculate the loss between the predicted action $A'$ and the ground truth action $A$. The total loss of PIVOT-R is $\mathcal{L} = \mathcal{L}_{\text{scene}} + \mathcal{L}_{\text{act}}$.

# 4 Experiments

We conduct experiments on the challenging SeaWave [42] benchmark. Our experiments aim to address three key inquiries: 1) How effective is PIVOT-R in executing various complex language instructions? 2) How robust and generalizable is PIVOT-R to manipulation on out-of-distribution scenarios? 3) Which modules of PIVOT-R play an important role? 4) if there are cases where the robot can retry and successfully perform an action after an initial incorrect attempt?

## 4.1 Experiment Settings

- **AHE.** We use multithreading to process each module separately. Each thread runs at its own frequency, extracts the latest data from the corresponding buffer, and places the output results in the buffer. For example, the VLM gets data from the camera buffer and saves the output in the buffer after each update. Then, the scene and action prediction module updates at different

**Instruction: "adjust the position of the green bottle so that it is nearer to the blue one."**

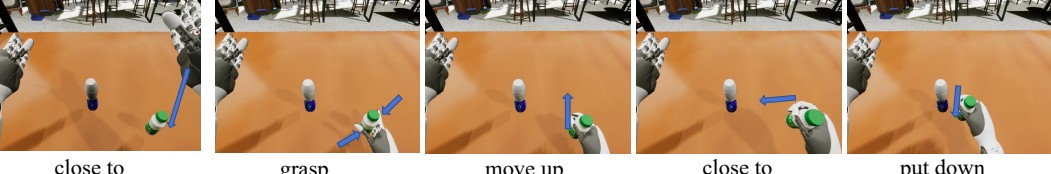

| close to | grasp | move up | close to | put down |
|---|---|---|---|---|

Figure 3: Examples show the execution process of PIVOT-R. The text below the image describes the primitive actions to be performed next. Blue arrows indicate the direction of actions.

frequencies and reads the latest data from the cache of the previous module. Different modules will not be blocked by other parts. For the execution frequency of different modules in AHE, we set $v_1 = 3, v_2 = 10, v_3 = 30$.

- **Primitive actions.** We divide primitive actions according to the object-centered principle, including "close to", "grasp", "move up", "move down", "release", "rotate + (direction)", "push + (direction)", "pull + (direction)", "open", and "close", a total of 10 types. For example, for the primitive action "close to", its text description is defined as "move close to the target object". More detailed information is presented in the Appendix A.1.

- **Action prediction.** For action prediction, PIVOT-R predicts delta XYZ positions and delta Euler angles for movement and binary state of the gripper. Similar to RT-1[7], we discretize each action dimension into 256 bins, ensuring that the value distribution of each bin is uniform.

More experiment settings are shown in Appendix A.2.

## 4.2 Benchmark and Baselines

**Simulation Benchmark.** We choose SeaWave [42], an open-source benchmark to learn multi-level instruction tasks, as our experimental platform, and use the corresponding data as demonstration data for imitation learning. Its greatest advantage is that it provides progressive tasks, facilitating our comprehensive comparison and analysis of the model's capabilities. It supports 8 skills, including daily operations such as grasping and placing objects, opening and closing doors, and more than 3,000 different instructions. The SeaWave dataset contains a total of 13K data covering four different levels of language instructions. We train on this dataset and test on a specially divided test set. A more detailed introduction is in Appendix B.

In addition, we added primitive action and waypoint annotations to the dataset. For ground truth waypoints $M$, we define the action frame that meets one of the following two conditions in data collection as the waypoint state of robot manipulation: 1) primitive action completion frame; 2) the speed of the robotic arm approaches zero or the state of the gripper changes. We annotate the frames that satisfy the conditions as waypoints $M$ along with the corresponding primitive actions.

**Real-world Evaluation.** We conducted real-world experiments, where we set up three tasks: (i) "Pick up": pick up the correct object from the table. (ii) "Put on": Pick up the object and place it on the correct color block. (iii) "Push to": Push the object to the correct color block. We collected 400, 200, and 200 sets of demonstrations respectively. We tested each task 24 times to calculate the average success rate.

**Baseline.** In the experiment, we selected BC-Z [23], Gato [41], RT-1 [7], Octo [36], GR-1 [55], and Surfer [42] as the baseline models for the SeaWave benchmark. BC-Z [23] includes a pre-trained multilingual sentence encoder, a FiLM encoder, and a two-layer MLP to decode robot actions. Gato [41], RT1 [7] and Octo [36] all embed text and images, and then use Transformer to output actions end-to-end. They are currently relatively simple and effective methods. SUSIE [6] predicts sub-goals through video predictors and GR-1 [55] enhances model effectiveness with video generation pre-training. By predicting the future and explicit modeling of the action and scene prediction, Surfer [42] achieved the SoTA performance on SeaWave with the same amount of data. We train these models on the full SeaWave dataset to allow for a fair comparison.

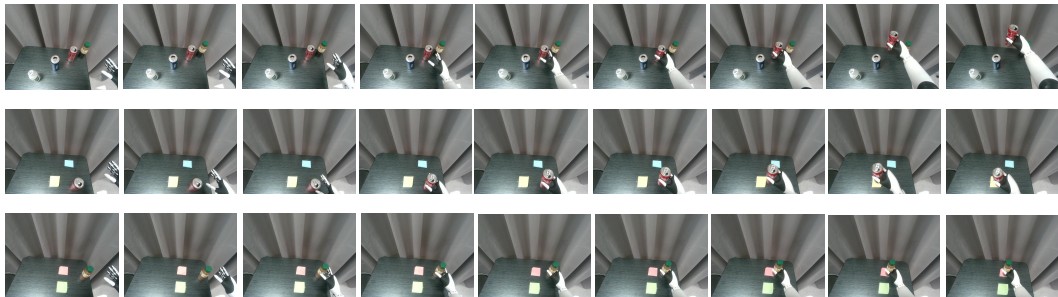

Figure 4: We show demonstrations of real world evaluation. The first row is "pick up the coke", the second row is "put the red bottle on the yellow block", and the third row is "push the object on the desk to the pink block".

## 4.3 Results on Robotic Manipulation

**Results on SeaWave.** We perform experiments on four levels of tasks in SeaWave, and the average success rate is in the last column. The results are shown in Table 1. PIVOT-R substantially achieved a significant improvement on all tasks. Specifically, PIVOT-R achieved an average success rate of 74.19%, 19.45% higher than the best baseline. Both the manipulation ability and the ability to understand instructions have been greatly improved. This confirms the effectiveness of the primitive-driven approach.

We also show qualitative results, which are shown in Figure 3. It demonstrates the example of bringing milk close to yogurt. The task process can be divided into five actions. Through the instruction of primitive actions and the prediction of waypoints, the model successfully completes the task.

It is also important for robots to be able to operate in real-time. Since the hardware device and action space are the same for all models, we focus on the inference speed of the models. As shown in the last column of Table 1, we compare the inference time of the models. We calculated the average time for the model to execute one step. It can be seen that BC-Z based on ResNet[19] is the fastest. In addition, the inference speed of PIVOT-R and most other models are of the same order of magnitude, with only a few milliseconds difference. Though simple, AHE's integration with WAWM is highly effective. PIVOT-R's VLM-based primitive-driven WAWM for scene and action prediction, combined with AHE for asynchronous execution, improves efficiency by 28 times.

**Results on Real World.** The quantitative results are shown in Table 2. PIVOT-R improves the average success rate by 6%. The qualitative results are shown in Figure 4. Surfer and RT-1 usually fail due to position errors, while PIVOT-R has higher accuracy. In the "push to" task, the performance of all models is suboptimal. This is because the downward force applied during the pushing process increases the resistance, making it difficult for the models to effectively predict and adapt to this change.

Table 2: Performance of different methods on three real robot manipulation tasks (%). "Pick up": pick up the correct object from the table. "Put on": Pick up the object and place it on the correct color block. "Push to": Push the object to the correct color block.

| Model | Pick up | Put on | Push to | Mean |
|---|---|---|---|---|
| Octo | 34.72 | 27.78 | 4.17 | 22.22 |
| RT-1 | 40.28 | 22.22 | 19.44 | 27.31 |
| GR-1 | 26.39 | 29.17 | 8.33 | 21.30 |
| Surfer | 41.67 | 29.17 | **31.94** | 34.26 |
| PIVOT-R | **54.17** | **41.67** | 25.00 | **40.28** |

## 4.4 Generalization Ability

We also perform experiments in different unseen scenarios on level 2, 3, and 4 tasks. New scenarios include unseen backgrounds (*i.e.*, two unseen tables), changing light intensity, and more distractions (*i.e.*, more objects). The results are shown in Table 3. PIVOT-R still maintains a success rate far superior to other models, indicating that with the help of WAWM, the model captures key information and maintains good generalization in changing scenarios.

Table 3: Performance comparison on seen scenarios, different backgrounds, changing lights, and more distractors (%).

| Model | Seen | Unseen backgrounds | Changing lights | Distractors |
|---|---|---|---|---|
| Gato [41] | 24.56 | 20.83 | 23.33 | 16.67 |
| BC-Z [23] | 27.02 | 19.17 | 18.33 | 21.67 |
| Octo [36] | 38.92 | 40.83 | 37.50 | 35.83 |
| RT-1 [7] | 41.05 | 38.33 | 40.83 | 35.00 |
| GR-1 [55] | 42.40 | 40.83 | 35.00 | 37.50 |
| Surfer [42] | 48.07 | 46.67 | 45.83 | 40.83 |
| PIVOT-R | **69.57** (21.0 ↑) | **59.17** (12.5 ↑) | **61.67** (15.84 ↑) | **55.83** (15.0 ↑) |

Table 4: Ablations studies of PIVOT-R in four levels of manipulation tasks (%).

| Model | Level 1 | Level 2 | Level 3 | Level 4 | Mean | Time(ms) |
|---|---|---|---|---|---|---|
| PIVOT-R | **88.06** | **77.55** | **73.33** | **57.82** | **74.19** | 27 |
| PIVOT w/ PAC | 82.22 | 73.33 | 69.17 | 51.67 | 69.10 ($-$**5.09**) | - |
| PIVOT w/ RSC | 72.92 | 40.83 | 35.83 | 25.00 | 43.65 ($-$**30.54**) | - |
| PIVOT-R w/ next frame | 72.92 | 45.92 | 34.33 | 24.63 | 44.45($-$**29.7**) | - |
| PIVOT-R w/ interval frame | 78.13 | 51.02 | 40.30 | 24.63 | 48.52($-$**25.7**) | - |
| PIVOT-R w/ final frame | 89.58 | 66.33 | 49.25 | 40.30 | 61.36($-$**12.8**) | - |
| PIVOT-R w/ Qwen-VL | 88.54 | 76.53 | 72.26 | 55.78 | 73.28($-$**0.9**) | - |
| PIVOT-R w/ GPT-4 | 87.50 | 78.57 | 74.45 | 59.18 | 74.92($+$**0.7**) | - |
| PIVOT-R w/o AHE | 90.63 | 80.61 | 76.64 | 60.54 | 77.11($+$**2.9**) | 756 |
| PIVOT-R w/ video decoder | 85.42 | 70.83 | 62.77 | 46.26 | 66.32($-$**7.8**) | - |
| PIVOT-R w/ large action module | 87.50 | 75.51 | 68.61 | 53.28 | 71.23($-$**2.9**) | - |

## 4.5 Ablation Study

In this section, we explore what is important in the design of the model. Specifically, We discuss the impact of waypoint selection, VLM, AHE, scene prediction supervision, and action prediction module design on PIVOT-R's performance. We designed a series of ablation experiments. We made some assumptions and experiments: *(i)* Waypoint selection. We conduct experiments by selecting the primitive action completion (PAC) frame, robot state changes (RSC) frame, next frame, five frames apart, and the final frame of the trajectory as waypoints respectively. *(ii)* VLM's image and language understanding capabilities. We chose Qwen-VL[3] of the same size to compare with the most powerful GPT-4[37] currently. *(iii)* Design of asynchronous architecture. We canceled the asynchronous architecture so that each module will be updated at every step. *(iv)* Design of scene prediction module. We refer to the design of MAE[18] and use predicted pixel-level images instead of feature prediction. *(v)* Design of action prediction module. We use a larger Transformer. Table 4 shows the results of each ablation and the delta performance compared.

**Waypoint selection.** As shown in the results, the performance of PIVOT-R with only primitive action completion frames dropped by 5.1%, the performance of PIVOT-R with robot state change frames dropped by 30.54%. Therefore, action completion frames are the main contributing factor. Selecting the next frame or selecting the interval frame both caused a significant drop in performance, indicating that the waypoint information for these two choices was too little or confusing. The performance of selecting the final frame as a waypoint also dropped a lot, indicating that it is an effective method to guide the model according to primitive actions.

**VLM selection.** Different VLM models, whether they are models of the same level or the current largest and best models, do not bring significant performance changes. This shows that our method does not strongly depend on VLM. PIVOT-R gives full play to the understanding and reasoning capabilities of VLM and makes up for the shortcomings of VLM in the dynamic world in the scene prediction module.

**Other designs.** Changing the model to a synchronous serial structure has some improvements (2.9 ↑), but it's 30 times slower. Considering the requirements of real-time operation, we use an asynchronous parallel architecture, taking into account both success rate and speed. We also discussed the design of the scene prediction module. Compared with the original prediction at the high-level feature level, pixel-level prediction caused a decrease in performance. We suspect that this is because pixel-level

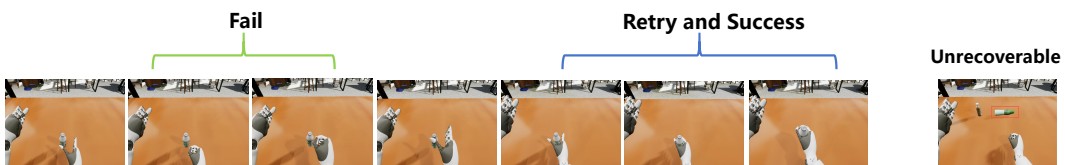

Figure 5: Left: Example of retry and successful execution after a manipulation error occurred. Right: Example of retry still failing. After an object is knocked down and rolled a certain distance, it is difficult to successfully grab it again.

prediction focuses too much on detailed information, causing key information to be ignored. Finally, we also test whether the design of the current action prediction module is reasonable and whether the action prediction module needs a larger model to make better predictions. Experiments have shown that the action prediction module only needs a small model to complete the task well, but a larger model may cause over-fitting.

### 4.6 Failure and Retry

This section discusses cases where the robot fails and whether it can be retried and successfully executed. As shown in Figure 5 (left), retries may be successful in some cases. When the position of the gripper deviated and the object failed to be grasped, the second attempt to grasp was successful. However, in the case of Figure 5 (right), if the object is knocked down and rolls a certain distance, it will be difficult to successfully grasp it again.

## 5 Discussions

**Conclusion.** In this paper, we propose PIVOT-R, a primitive-driven waypoint-aware world model. PIVOT-R focuses on the execution of primitive actions. Predicting key waypoints in the future greatly improves performance. It has achieved state-of-the-art results on the SeaWave benchmark, and experiments have proven that it has good robustness. We also use asynchronous hierarchical executors to ensure fast enough execution of the model. In addition, we demonstrate that PIVOT-R has the potential to complete unseen instructions and tasks under the guidance of a high-level VLM. Finally, we also demonstrate PIVOT-R's ability to improve performance through human demonstration. These results illustrate the potential of PIVOT-R.

**Limitations.** We demonstrate the ability of PIVOT-R to complete tasks, even unseen tasks, through a combination of primitive actions guided by instructions. However, action execution and instructions are sometimes inconsistent. For example, if "push left" is required, the robot may execute "push front". Therefore, we also need to strengthen the consistency between high-level instructions and underlying actions, so that the robot can truly perform tasks according to our instructions, and even adjust according to requirements, just like a real intelligent agent.

## 6 Acknowledgement

This work was supported in part by National Science Foundation of China Grant No.62476293, Guangdong Outstanding Youth Fund (Grant No. 2021B1515020061), Shenzhen Science and Technology Program (Grant No. GJHZ20220913142600001), Nansha Key RD Program under Grant No.2022ZD014, The Major Key Project of PCL (No. PCL2024A04, No. PCL2023AS203).

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

**SUMMARY OF THE APPENDIX**

This appendix contains additional details for this paper. The appendix is organized as follows:

- §A provides **Experiment Details**.
- §B shows more **SeaWave Benchmark Details**.
- §C shows some **Emergent Capabilities** of **PIVOT-R**.
- §D shows **More Experiments**.
- §E shows **More Results**.
- §F lists **Prompt Details** used in experiments.

# A    Experiment Details

## A.1    Primitive Actions

Table 5: The defined primitive actions and their textual descriptions.

| Primitive Action | Description |
|---|---|
| Close to | Move close to the target object |
| Grasp | Hold or pick up the target object |
| Move Up | Lift the target object upwards |
| Move Down | Lower the target object downwards |
| Release | Let go of or put down the target object |
| Rotate + (direction) | Turn the target object |
| Push + (direction) | Push the target object |
| Pull + (direction) | Pull the target object |
| Open | Open an object, such as a door or container |
| Close | Close an object, such as a door or container |

## A.2    Training Details

All experiments involved in this paper are conducted on a single GPU server with 6 NVIDIA RTX-4090 GPUs. PIVOT-R selects LLAVA[31] as the high-level VLM and selects Transformers of 12 layers and 3 layers as the scene prediction module and action prediction module respectively. We froze VLM and encoder, and PIVOT-R has trainable parameters of 30 M in total. The hyperparameter settings for PIVOT-R are shown in Table 6.

Table 6: The hyperparameter setting of Imitation Learning.

| Hyperparameters | Value |
|---|---|
| LS | 12 |
| LA | 3 |
| Image encoder | CLIP-ViT-B/32 |
| Text encoder | CLIP-ViT-B/32 |
| Transformers heads | 8 |
| Embedded dims | 512 |
| Learning rate | 3e-5 |
| dropout | 0.1 |

# B    SeaWave Benchmark Details

In order to meet the needs of common robot operations, SeaWave has designed 8 skills, the detailed definitions are shown in Table 7. And SeaWave proposes progressive tasks. Natural language is one of the most direct and effective ways of human-computer interaction. However, due to the complexity and variability of external visual scenes and human natural language instructions, understanding

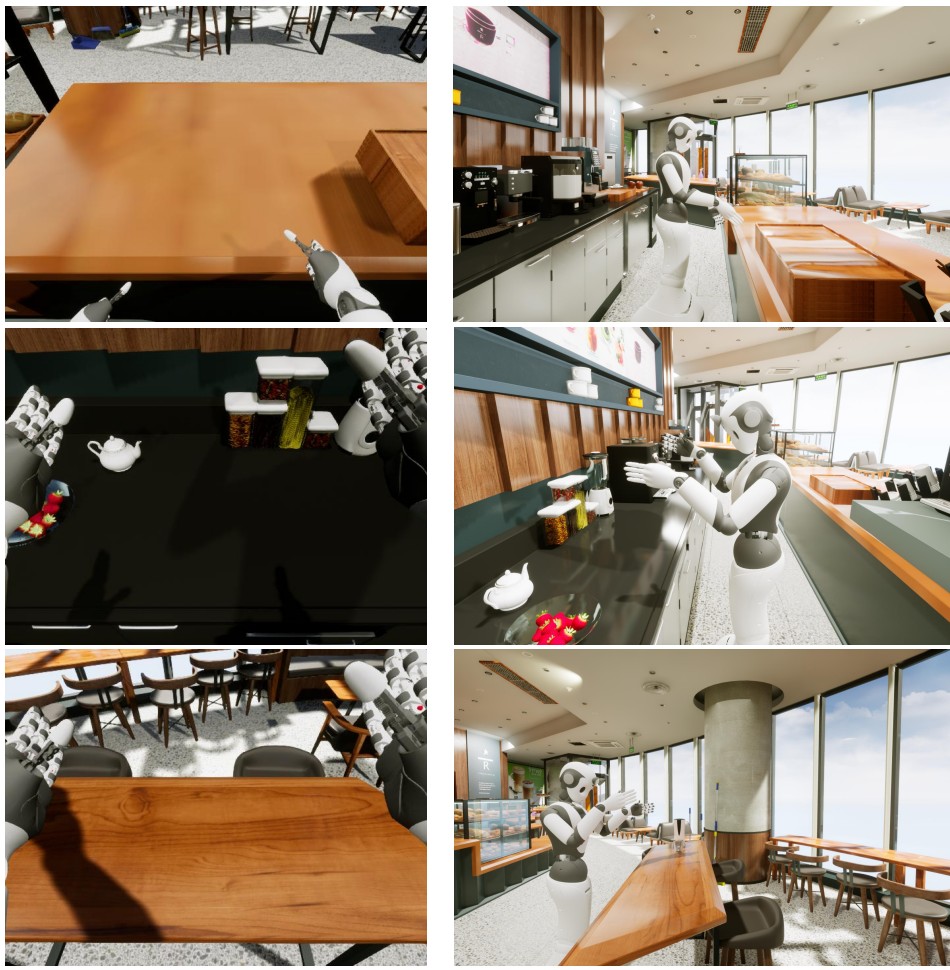

Figure 6: **Scenes in SeaWave.** The left column represents the first-perspective image, and the right column represents the third-perspective image. The scenes in the same row are the same.

and executing these instructions has become one of the key challenges in embodied AI research. To systematically analyze and study these challenges, SeaWave classified tasks into four levels according to the complexity of instructions and the ease of operation. Task definition and examples are described in Table 8. The specific content is as follows:

- **Level 1:** The scene contains only one object, and the robot receives explicit machine language commands consisting of *verbs + nouns*. It is used to evaluate the basic manipulation capabilities of the model.

- **Level 2:** This task scenario contains multiple objects and the natural language instructions explicitly include the name of the target object. It is used to evaluate the model's ability to understand conventional natural language instructions.

- **Level 3:** This task scene contains multiple objects, but the natural language instructions do not contain the name of the target object, but only provide expressions related to the functionality of the target object. It is used to evaluate the model's ability to infer the intent of human instructions.

- **Level 4:** This task scene contains multiple objects. The natural language instructions do not include the name of the target object but only provide expressions related to the function, appearance, or location of the target object. This instruction requires the model to have strong visual and language information processing capabilities at the same time. It is used to evaluate the model's visual perception and decision-making capabilities.

Table 7: The list of skills on SeaWave and their description and success condition.

| Skill | Description | Success Condition |
|---|---|---|
| Pick Target | Grasp the target object and lift it | The target object is 10cm away from the table |
| Place Target | place the target object on the table | The target object stands upright on table |
| Move A Near B | Grasp A and move it closer to B | A is moved and ends up being less than 10cm away from B |
| Open Door | Open the door | The door is opened more than 80 degrees |
| Close Door | Close the door | The door is closed to less than 10 degrees |
| Push Target Front | Push the target object forward | The target object is pushed forward 10cm |
| Push Target Aside | Push the target object to the left or right | The target object moves 10cm to the left or right |
| Knock Target Over | Knock the target object over | The target object falls on the table |

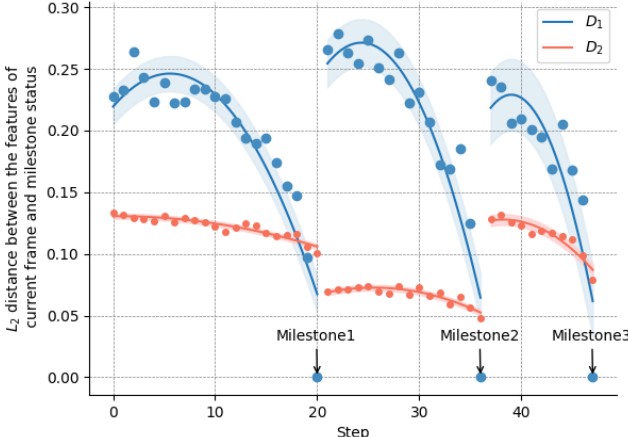

Figure 7: Feature analysis. The blue points $D_1$ represent the distance between $F_{O_t}$ and $F_{M_t}$ in the spatial dimension, and the red points $D_2$ represent the distance between $F_{M'_t}$ and $F_{M_t}$ in the spatial dimension. We fit curves to these points and draw confidence intervals for better observation.

We also show some scenes in SeaWave in Figure 6. SeaWave is a highly simulated scene built based on UE5 and its scenes are realistic.

Table 8: The setting of progressive reasoning tasks.

| Level | Single/Multiple Objects | Capability Assessment | Example |
|---|---|---|---|
| 1 | Single | Basic manipulation | pick the milk |
| 2 | Multiple | Natural language understanding | can you please take the gluestick off the table |
| 3 | Multiple | Intention inference | could you please go grab a refreshing beverage for me |
| 4 | Multiple | Visual perception and decision-making | retrieve the object located behind the one to the right |

## C   Extra Studies

In this section, we analyze why PIVOT-R succeeds, exploring its generalization to new tasks and potential for further improvement by incorporating other datasets.

### C.1   Feature Analysis

In this study, we performed an in-depth comparative analysis. We define that $F_{O_t}$, $F_{M'_t}$, and $F_{M_t}$ represent the features of $O_t$, $M'_t$, and $M_t$ respectively and try to explore the spatial distance relationships between the feature $F_{M't}$ (red points) predicted by PIVOT-R and the real-time observed feature $F_{O_t}$ (blue points) relative to the feature $F_{M_t}$. As shown in Fig 7, the $L_2$ distance between $F_{O_t}$ and $F_{M_t}$ gradually decreases as the task progresses, a phenomenon that is critical to the functionality of the action execution module. The main task of the action execution module is to adjust the action so that $O_t$ moves closer to $M_t$, thereby reaching the target state. $F_{M'_t}$ (red dots) provides significant guidance for action prediction. $F_{M'_t}$ not only predict the possible locations of $F_{M_t}$ in space but also show smaller variances in long-term series analysis, indicating that their predictions are more stable, thus greatly improving the accuracy and reliability of model manipulation.

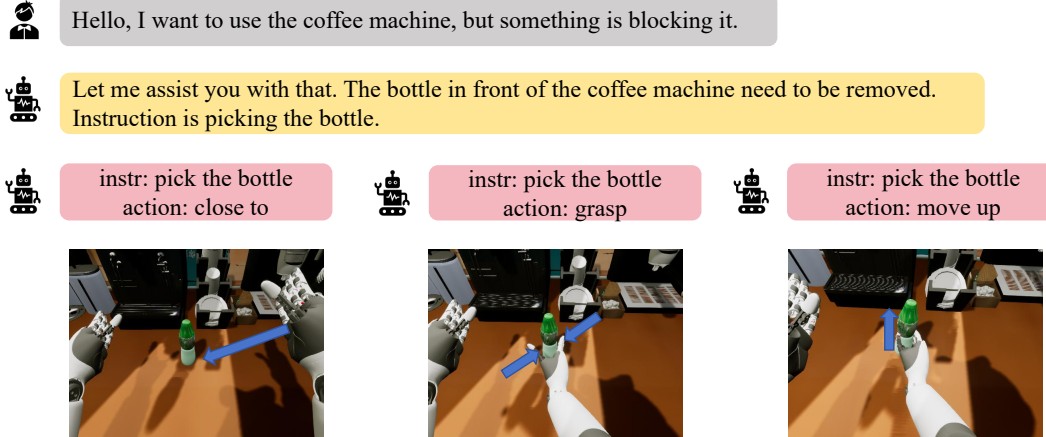

Figure 8: An example shows that when an out-of-distribution instruction is encountered, the VLM's reasoning ability is used to parse it into a learned task instruction, allowing the model to successfully complete the task.

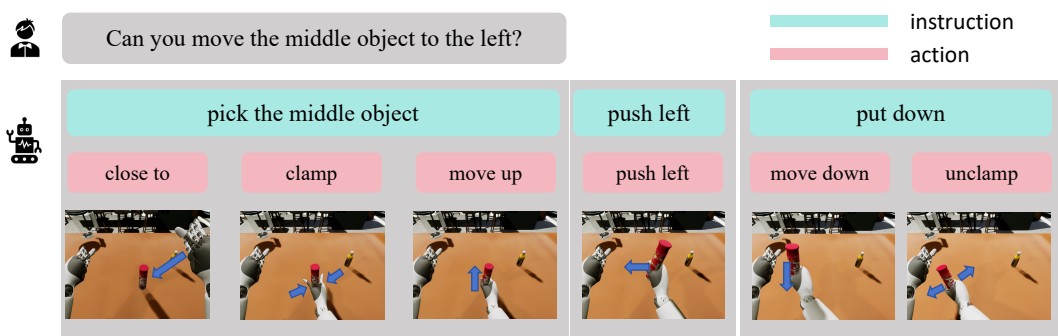

Figure 9: An example shows that for an unseen task, based on the skills and actions that have been learned, PIVOT-R can break down the task and combine the actions to complete the task.

## C.2 Emergent Capabilities

### C.2.1 Generalization to Out-of-distribution Instructions

In this section, we explore whether PIVOT-R can use the reasoning capabilities of VLM to understand out-of-distribution instructions. Although the model has only received instructions from the SeaWave training set, we can use VLM to parse the instructions into learned instructions, so that PIVOT-R can understand and execute out-of-distribution instructions. To this end, we designed a prompt to let VLM analyze and process the new instructions. The details of the prompt are shown in the Section F.2.

We show an example, as shown in Figure 8, We qualitatively observe that for the instruction "Hello, I want to use the coffee machine, but something is blocking it.", VLM infers that the task that needs to be performed is "remove the bottle in front of the coffee machine.", and replaces the original instruction to the learned form "pick the bottle". At this point, PIVOT-R can complete the task based on the skills it has learned.

### C.2.2 Generalization to New Tasks

Zero-shot unseen tasks generalization is very difficult. Nevertheless, we hope to prove that PIVOT-R can complete new tasks through the combination of existing primitive actions, because the primitive actions are shared between different tasks. This requires some appropriate adjustments. To this end, we provide a new prompt to let VLM assist in completing this work. The details of the prompt are shown in the Section F.3.

Table 9: Experimental results for training with additional human data.

| Model | Seen | Unseen backgrounds | Changing lights | Distractors |
|---|---|---|---|---|
| Origin | **69.57** | 59.17 | **61.67** | 55.83 |
| Co-Train | 65.75 | 60.83 | 56.67 | 52.5 |
| Pre-Train | 67.78 | **63.33** | 58.83 | **58.83** |

As shown in Figure 9, with the help of VLM, the new task is decomposed into existing primitive actions. Specifically, for the instruction "Can you move the middle object to the left?", VLM first decomposed it from the instruction level into "pick", "push left", and "put down", and then further decomposed it into the learned primitive actions. In the end, PIVOT-R completed the task according to the guidance of VLM.

Zero-shot unseen instruction and task generalization are very difficult. Nevertheless, we hope to prove that PIVOT-R can complete new tasks through VLM guidance and the combination of learned actions. Although the tasks are different, their primitive actions are shared. For example, for the unseen task "move the middle object to the left", VLM first decomposed it into the learned primitive actions "close to", "clamp", "move up", "push left", "put down" and "unclamp". Finally, PIVOT-R completed the task according to the guidance of VLM. More details are shown in Appendix C.2.

### C.2.3 Train with Human Data

We also explored PIVOT-R's ability to utilize other data. Embodied AI has been limited by a lack of robot data. We see if we can use other data to enhance the model. Although some data do not contain robot actions, they are still helpful for training our scene prediction module. To do this, We use the Ego4D dataset, which is a large-scale first-person perspective video dataset. It contains more than 3,500 hours of data, and each video clip contains detailed annotation information to describe human behavior. We train based on the benchmark data of the "Short Term Object Interaction Anticipation Challenge", which aims to predict the next human-object interaction happening after a given timestamp. Each piece of data contains a 0.25s~1s video and the corresponding operation objects and operation action, which exactly meets the input and output label requirements of the scene prediction module.

Specifically, for Ego4D data, the scene prediction module accepts the input of the initial frame and the current action instruction, outputs the features of the predicted frame, and calculates the loss with the features of the annotated end frame. We use two training methods, one is to mix Ego4D and SeaWave data for co-training, and the other is to use Ego4D for pre-training first, and then use SeaWave data for fine-tuning.

As shown in Table 9, we compared co-training and pre-training results. It can be seen that co-training does not bring better results. We guess it is because the data distribution is very different, making it difficult to train the model. Although the success rate of Pre-training has dropped slightly in seen scenarios, it has improved significantly in unseen backgrounds and more distractors scenarios, increasing by 4.16% and 3.00% respectively, indicating that PIVOT-R can make use of other data to improve the generalization ability.

## D More Experiments

We evaluated PIVOT-R on the latest SIMPLER [28] benchmark, a scalable, repeatable and reliable proxy for real-world evaluation. We use this to verify the scalability of PIVOT-R in the real world. As shown in Table 10, PIVOT-R outperformed the best baseline by nearly 10%.

Table 10: Performance comparison of different methods on SIMPLER benchmark (%). SIMPLER is a simulation benchmark which evaluation can be a scalable, reproducible, and reliable proxy for real-world evaluation. It selects four tasks from Bridgedata.

| Model | Put Spoon on Towel | Put Carrot on Plate | Stack Green Blockon Yellow Block | Put Eggplant in Yellow Basket | Mean |
|---|---|---|---|---|---|
| RT-1-X | 0.000 | 0.042 | 0.000 | 0.000 | 0.011 |
| Octo-Base | 0.125 | 0.083 | 0.000 | 0.431 | 0.160 |
| Octo-Small | **0.472** | 0.097 | 0.042 | 0.569 | 0.295 |
| PIVOT-R | 0.417 | **0.278** | 0.000 | **0.875** | **0.393** |

# E  More Results

More examples are shown in Figure 10, covering tasks at various levels.

# F  Prompt Details

## F.1  Prompt Details for Primitive Action Parsing

To ensure that the Vision Language Model (VLM) produces text that adheres to our criteria, we have meticulously crafted a multi-stage dialogue process, complemented by comprehensive prompts. The procedure unfolds as follows: initially, we prompt the VLM to depict the scenario; subsequently, the VLM specifies the actions that need to be undertaken; and ultimately, the VLM determines the action to be executed at the current juncture.

We have a total of three rounds of dialogue. The following are the prompts for each round of dialogue.

1. **Describe the scene.**

> Given a task, which is for a mobile Franka panda robotic arm to learn a manipulation skill in the simulator.
> Your task is to help me break down the process of the robot performing the task into several actions to help the robot better understand and execute.
> Capabilities: The task can only be completed with a robotic arm, which can move, rotate and clamp.
>
> You should output the response using the same format as the following:
> ```
> """
>     "scene": "You should description the scene"
> """
> ```
>
> Here is one example:
> ```
> """
>     Input: Close the red jar.
>
>     Output: On the table, there is a red jar, a blue jar, and a
>       bottle cap
> """
> ```
>
> Can you do it for the following input:
> ```
> """
>     Task: {task}
> """
> ```

2. **Imagine the actions that need to be done.**

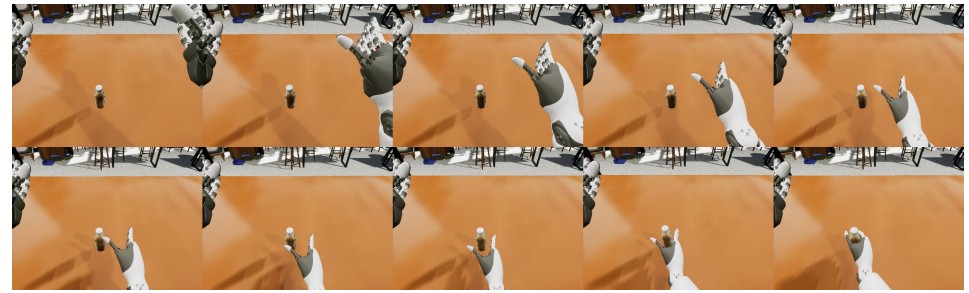

**Level 1: "pick the coffee"**

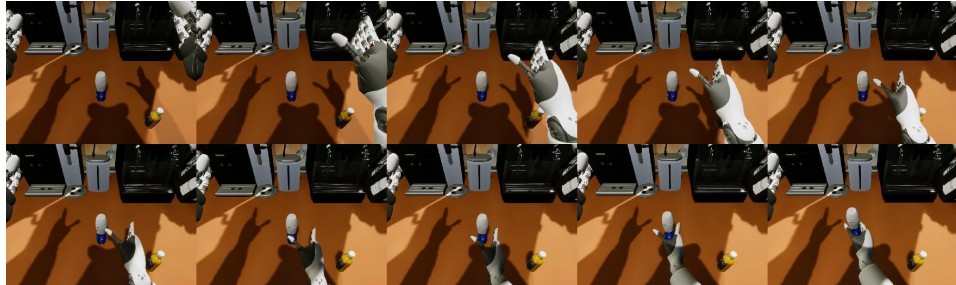

**Level 2: "Can you get me a bottle of yogurt?"**

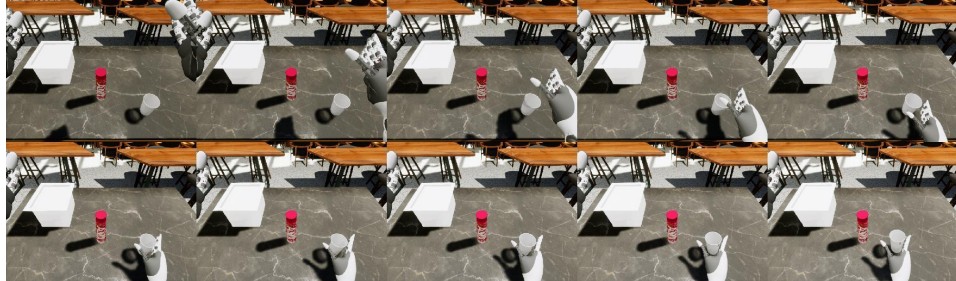

**Level 3: "I need a container. Can you move it to the left?"**

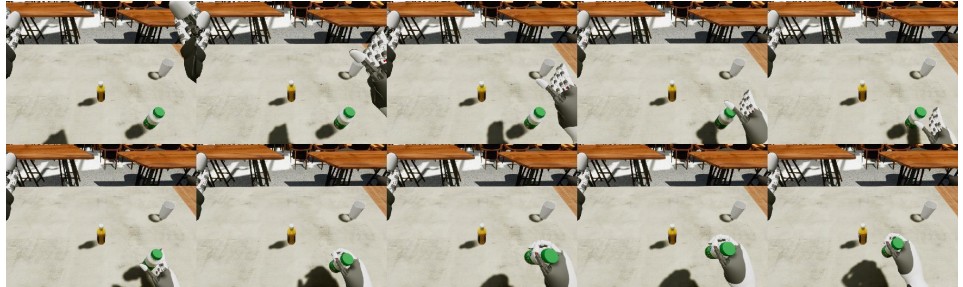

**Level 4: "Can you move the green object closer to the one on its left?"**

Figure 10: Rollouts on multi-level tasks of the SeaWave benchmark.

Given a task, which is for a mobile Franka panda robotic arm to learn a manipulation skill in the simulator. Your task is to help me break down the process of the robot performing the task into several actions to help the robot better understand and execute.
Capabilities: The task can only be completed with a robotic arm, which can move, rotate and clamp.

You should output response using the same format as the following:

```
"""
    "actions": [
```

```
        {
            "action": "The action name",
            "target": "The target object"
        },
        ... #actions which are needed to complete the task
    ]
"""
```

The actions you can choose include the following:

```
"""
    move to : move the gripper closer to an object,

    clamp : use gripper to clamp the object,

    unclamp : open gripper to unclamp the object,

    screw : rotate the gripper for opening or closing lid,

    lift : lift the object,

    push : push the object + (direction),

    pull : pull the object + (direction),
"""
```

Here is one example:

```
"""
Input:
Task: close the red jar.
Scene: On the table, there is a red jar, a blue jar, and a bottle
  cap.

Output:
    "actions": [
        {
            "action": "move to",
            "target": "the bottle cap"
        },
        {
            "action": "clamp",
            "target": "the bottle cap"
        },
        {
            "action": "move to",
            "target": "the red jar"
        },
        {
            "action": "rotate",
            "target": "the bottle cap"
        }
        ]
}
"""
```

Can you do it for the following task:

```
"""
    Task: {task}
```

```
    Scene: {scene}
"""
```

3. **Decide what action to take now.**

Given a task, which is for a mobile Franka panda robotic arm to learn a manipulation skill in the simulator. Your task is to help me break down the process of the robot performing the task into several actions to help the robot better understand and execute.
Capabilities: The task can only be completed with a robotic arm, which can move, rotate and clamp.

You should output one action that should be done at the current moment, and only can output one. You should output response using the same format as the following json file, and don't need to output escape characters

```
"""
{
    "do_action" {
            "action": "The action name",
            "target": "The target object"
        }
}
"""
```

The actions you can choose include the following:

```
"""
    move to : move the gripper closer to an object,
    clamp : use gripper to clamp the object,
    unclamp : open gripper to unclamp the object,
    screw : rotate the gripper for opening or closing lid,
    lift : lift the object,
    push : push the object + (direction),
    pull : pull the object + (direction),
"""
```

Here is one example:

```
"""
Input:
Task: close the red jar.
Scene: On the table, there is a red jar, a blue jar, and a bottle
  cap.
Actions: [
        {
            "action": "move to",
            "target": "the bottle cap"
        },
        {
            "action": "clamp",
            "target": "the bottle cap"
        },
        {
            "action": "move to",
            "target": "the red jar"
        },
        {
            "action": "rotate",
```

```
                    "target": "the bottle cap"
                }
            ]

    Output:
    {
        "do_action":
            {
                "action": "move to",
                "target": "the red jar"
            }
    }
    """
```

Can you do it for the following task:

```
    """
        Task: {task}
        Scene: {scene}
        Actions: {actions}
    """
```

## F.2 Prompt Details for New Instructions

In order to be able to process out-of-distribution instructions, we let VLM process the commands first and parse them into learned tasks. To do this, we set a prompt as shown below.

Given a task, which is for a mobile Franka panda robotic arm to learn a manipulation skill in the simulator. Your task is to help me break down the process of the robot performing the task into several actions to help the robot better understand and execute.
Capabilities: The task can only be completed with a robotic arm, which can move, rotate and clamp.

You need to give an instruction base on the skills you have learned according to the given tasks. You should output the response using the same format as the following json file:

```
"""
{
    "instruction": "You should description the instruction here",
}
"""
```

The skills you have learned:

```
"""
    Pick Target: Grasp the target object and lift it,
    Place Target : place the target object on the table,
    Move A Near B : Grasp A and move it closer to B,
    Open Door : Open the door,
    Close Door : Close the door,
    Push Target : push the object + (direction),
    Knock Target Over : Knock the target object over,
"""
```

Here is one example:

```
"""
Input:
Task: The bottle is on the edge of the table, it's too dangerous.
```

```
Output:
{
    "instruction": "Push the bottle front",
}
"""
```

Can you do it for the following task:

```
"""
    Task: {task}
"""
```

## F.3 Prompt Details for New Tasks

In order to be able to solve new tasks, we let VLM process the commands first and parse them into learned tasks and actions. To do this, we set a prompt as shown below.

Given a task, which is for a mobile Franka panda robotic arm to learn a manipulation skill in the simulator. Your task is to help me break down the process of the robot performing the task into several actions to help the robot better understand and execute.
Capabilities: The task can only be completed with a robotic arm, which can move, rotate and clamp.

You need to complete a given task, based on the skills and actions you have learned. You should output the response using the same format as the following json file:
```
"""
{
    "instruction": "You should description the instruction here",
    "actions": [
        {
            "action": "The action name",
            "target": "The target object"
        },
        ... # actions which are needed to complete the task
    ]
    "do_action" {
            "action": "The action name",
            "target": "The target object"
        }
}
"""
```

The skills you have learned:
```
"""
    Pick Target: Grasp the target object and lift it,
    Place Target : place the target object on the table,
    Move A Near B : Grasp A and move it closer to B,
    Open Door : Open the door,
    Close Door : Close the door,
    Push Target : push the object + (direction),
    Knock Target Over : Knock the target object over,
"""
```

The actions you can choose include the following:
```
"""
    move to : move the gripper closer to an object,
```

```
    clamp : use gripper to clamp the object,
    unclamp : open gripper to unclamp the object,
    screw : rotate the gripper for opening or closing lid,
    lift : lift the object,
    move : move the object + (direction),
"""
```

Here is one example:

```
"""
Input:
Task: Hello, I'm on your right, can you bring me the object on the
  table.

Output:
{
    "instruction": "Pick up the object and move right",
    "actions": [
        {
            "action": "close to",
            "target": "the object"
        },
        {
            "action": "clamp",
            "target": "the object"
        },
        {
            "action": "move up",
            "target": "the object"
        },
        {
            "action": "move right",
            "target": "the object"
        }
    ],
    "do_action":
        {
            "action": "close to",
            "target": "the object"
        }
}
"""
```

Can you do it for the following task:

```
"""
    Task: {task}
"""
```

