# OpenReview forum: "PIVOT-R: Primitive-Driven Waypoint-Aware World Model for Robotic Manipulation"
_NeurIPS.cc/2024/Conference — NeurIPS 2024 poster_

### Official Review · Reviewer_9FHi · 2024-06-18

**Soundness:** 3
**Presentation:** 2
**Contribution:** 2
**Rating:** 5
**Confidence:** 4

**Summary:**

The paper introduces a Primitive-Driven Waypoint-Aware model for robotic manipulation tasks. Initially, the entire language instruction is broken down into specific sub-steps using VLM. Subsequently, each sub-step's corresponding feature ("waypoint") is predicted through image prediction to segment the entire task execution process. The article also employs different execution frequencies for different prediction steps to enhance the inference speed. Comparative experiments were conducted on the SeaWave benchmark, and the model's generalization ability in unseen scenarios and its learning capability from human videos were analyzed.

**Strengths:**

The figures and tables are clear and easy to understand. The experimental section reproduces several renowned studies and tests them on a new benchmark. The ablation study is elaborate.

**Weaknesses:**

1. The Introduction section contains misleading information and overclaims regarding the motivation. The model proposed in this paper has limited relation to a “world model”. I think that only the first step of parsing the language instruction has utilized the VLM, which makes it hard to convince that the entire model can be called a "world model". However, almost the entire article emphasizes the content related to the world model. Additionally, the paper points out that a significant drawback of previous works is sequential execution, and proposes the idea of accelerating inference speed for this reason. But the method of this paper is still sequential execution, and the acceleration is merely achieved by artificially setting different execution frequencies for different reasoning steps, which is also hard to consider as a strong contribution of this paper.

2. The method proposed in the article seriously lacks novelty, and the discussion of its relationship with related paper is also insufficient. I summarize that the article mainly proposes two methods: First, it decomposes language into multiple sub-goals, which the author refers to as waypoints. Similar methods were first proposed in SayCan, and the language parsing process in this paper is not learnable. Second, the article proposes predicting future waypoints as intermediate results. Similar future prediction methods have already been numerous, and related articles have also studied predicting future images (the approach of this paper) or the trajectory of future key points as different forms of sub-goals. Therefore, the technical contribution and innovation of this paper are quite lacking.

3. The experimental results of this paper are not very convincing.

-  a) The method of this paper has only been verified on a relatively limited SeaWave Benchmark. Moreover, for the method proposed in this paper, it is necessary to artificially process the data, such as limiting it to 10 primitive actions. And it is also necessary to add additional primitive action and waypoint annotations. This almost introduces additional annotation information that is not utilized in the baseline for comparison.

-  b) It is strange that this paper chooses Octo, RT-1, and GR-1 as baselines, as the training cost of these papers is very high, and it seems that this paper needs to re-train these models on new data, which requires a very long time for both training and testing. From the perspective of the method of this paper, it should be compared with some sub-goal prediction methods, such as UniPi and AVDC.

-  c) The results in Table 2 show that PIVOT-R's performance drops more in unseen scenarios compared to seen scenarios, how to explain this?

-  d) The experimental results in Table 4 have almost nothing to do with the motivation of the paper. The fact that pre-training with human data can improve the performance and generalization of downstream tasks has already been proven in a large number of previous works.

**Questions:**

My main concerns lie in the need for a clearer articulation of the motivation, a more explicit delineation of the contributions of this paper, and improvements to the experimental section.

**Limitations:**

The authors have adequately addressed the limitations and societal impacts.

---

> ### Author Rebuttal · Authors · 2024-08-07
>
> We are grateful to see your recognition of the PIVOT-R experiment. But there are some serious factual errors in the review:
>
> (1) VLM is **not equal to** the world model;
>
> (2) AHE is **multi-threaded** rather than sequential execution;
>
> (3) The task parsing of SayCan and PIVOT-R is NOT at the same level, and the learnability of language parsing is unnecessary;
>
> (4) Scene prediction is strongly related to the waypoint-aware world model and should not be viewed in isolation;
>
> (5) The experimental design and the motivation of PIVOT-R are strongly related and complete. We provide detailed responses to all the issues you have pointed out.
>
> We hope our response and rebuttal can address your concerns, and sincerely hope you can reconsider our work.
>
> **Q1.1.  About the world model.**
>
> VLM is not equal to the world model. The pioneering work of the world model [1] defines it as a model that can perceive the environment and predict changes in environmental states. In our PIVOT-R's waypoint-aware world model (WAWM), we use VLM to **perceive the scene and parse user instructions** to obtain primitive action descriptions. And we use the scene prediction module to **predict waypoint scenes** for capturing environmental dynamics. This fits the definition of a world model.
>
> [1] Ha, David, and Jürgen Schmidhuber. "World models." *arXiv preprint arXiv:1803.10122* (2018).
>
> **Q1.2. AHE's technical contribution.**
>
> PIVOT-R's AHE is implemented in a multi-threaded manner, with each module's output stored in a buffer to prevent blocks between modules. As shown in Table 3 in our paper, AHE improves PIVOT-R's execution efficiency by 28 times (27ms vs. 756ms). This approach is valuable for models with multiple modules of varying efficiencies, providing significant enhancement and useful reference for other work.
>
> **Q2. Waypoints and technical contributions.**
>
> **(a) Waypoints.**
>
> SayCan uses LLM to decompose complex human instructions into a higher-level sub-task, such as "find apples/go to table". This is significantly different from PIVOT-R, which is a decomposition of low-level primitive actions proposed for robot manipulation tasks.
>
> **(b)Technical contribution.**
>
> Waypoints' technical contributions should not be reviewed individually. PIVOT-R is the first attempt to introduce waypoint predictions for world model learning. It parses complex user instructions into primitive actions through VLM, and combines scene and action prediction modules driven by primitives to realize the modeling of physical world knowledge. This enables PIVOT-R to focus on task-relevant key points instead of being drowned in trivial scene predictions, leading to significant improvement in both manipulation performance and execution efficiency.
>
> **(c) Language parsing is not learnable.**
>
> As shown in the results of Lines 4 and 5 in Table 3 in our paper, the current open-source VLM is fully capable of handling the related primitive parsing tasks. Therefore, the introduction of VLM in the offline manner does not have a big impact on the whole performance.
>
> **Q3. Experiment.**
>
> **(a) Data annotation.**
>
> First, 10 primitive actions are sufficient for manipulation tasks. For example, [1,3] uses only pick and place actions, and ForceSight[2] defines 5 actions. Secondly, adding more annotation information to existing benchmarks is a common practice to improve model performance. Similarly, MOKA [4] and LEO [5] use LLM to annotate data for better performance. This annotation method itself is also one of the contributions of the work.
>
> [1] SHRIDHAR M, MANUELLI L, FOX D. CLIPort: What and Where Pathways for Robotic Manipulation[J].
>
> [2] COLLINS JeremyA, HOUFF C, TAN Y, et al. ForceSight: Text-Guided Mobile Manipulation with Visual-Force Goals[J]. 2023.
>
> [3] ZENG A, PETE F, TOMPSON J, et al. Transporter Networks: Rearranging the Visual World for Robotic Manipulation[J]. arXiv: Robotics,arXiv: Robotics, 2020.
>
> [4] Liu, Fangchen, et al. "Moka: Open-vocabulary robotic manipulation through mark-based visual prompting." *arXiv preprint arXiv:2403.03174* (2024).
> [5] Huang, Jiangyong, et al. "An embodied generalist agent in 3d world." *arXiv preprint arXiv:2311.12871* (2023).
>
> **(b) Reasons for choosing methods such as Octo and RT-1.**
>
> This is mainly because they are state-of-the-art robot manipulation models.
>
> **(c) Comparison with other sub-goal prediction methods.**
>
> We compared SUSIE [1], which uses InstructPix2Pix [2] for sub-object prediction, and low-level strategies for action prediction. As shown in Table 2 of the rebuttal Appendix PDF, PIVOT-R outperformed SUSIE by 26.93% (74.19% vs. 47.26%) on SeaWave tasks, demonstrating the effectiveness of WAWM in PIVOT-R for sub-goal modeling.
>
> [1] Black, Kevin, et al. "Zero-shot robotic manipulation with pretrained image-editing diffusion models." arXiv preprint arXiv:2310.10639 (2023).
>
> [2] Brooks, Tim, Aleksander Holynski, and Alexei A. Efros. "Instructpix2pix: Learning to follow image editing instructions." Proceedings of the IEEE/CVF Conference on Computer Vision and Pattern Recognition. 2023.
>
> **(d) Why does PIVOT-R show obvious performance degradation on unseen scenes?**
>
> This is normal for robot manipulation models, as they often struggle with generalization to unseen scenes not present in the training data, leading to perception or recognition errors. As shown in Table 2 in our paper, nearly all methods experience performance degradation in new scenarios. However, our model still achieves state-of-the-art performance.
>
> **(e) Train with Human Data.**
>
> This experiment is mainly to prove that PIVOT-R can also benefit from human data. We hope to conduct large-scale data training to improve model performance in the future. Based on your suggestion, we will move this experiment to the appendix in our revision.

---

> ### Author Response · Authors · 2024-08-12
> **Waiting for further discussion**
>
> Dear Reviewer,
>
> Thank you once again for your valuable feedback on our work. In our rebuttal, we have provided detailed responses to the weaknesses and questions you raised.  We have clarified the concept of world model, and the novelty of our waypoints and AHE. Additionally, we have elaborated on the experimental details, explaining data annotation and baseline selection, and added an additional sub-goal prediction baseline. We have also conducted additional experiments, including SIMPLER benchmark and real-world experiment to further illustrate the performance enhancements.
>
> As the author-reviewer discussion period is drawing to a close, we would like to check if there are any remaining questions or concerns that we can address. Please feel free to reach out if you need further clarification on any aspect of our work. We are committed to ensuring that our responses fully address your concerns and contribute to the improvement of our paper. We are confident that we have addressed all of your concerns and hope you will reconsider our work.

---

> > ### Comment · Reviewer_9FHi · 2024-08-12
> >
> > The author has addressed most of my doubts, and I have also seen that the author has provided additional experiments with other benchmarks as well as in real-world scenarios. I greatly appreciate the author's efforts. However, I still have some questions:
> >
> > 1. I understand that VLM is not equal to the world model, and I am also aware of the definition of the world model. But I still cannot fully comprehend the significance of the paper's repeated emphasis on the world model. As other reviewers have also pointed out, the method of this paper does not make a very clear and distinct differentiation from many other related works.
> >
> > 2. The author also mentioned that the method of this paper has incorporated more annotation information. Will the baseline method also introduce this information (considering the issue of fairness)? In addition, is the annotation process for this data quite laborious? For example, can it be efficiently generated through programs or LLMs, or does it require human intervention?
> >
> > 3. In my third point of weakness (c), what I mean is to focus on the performance degradation of each method after transferring to unseen conditions compared to its own seen conditions. It can be observed that PIVOT-R has dropped by about 10%, 8%, and 11% under three unseen conditions, respectively. While other baselines, such as Surfer, have dropped by 1.5%, 2%, and 7% under the three unseen conditions, respectively. Does this mean that PIVOT-R is more affected by environmental changes?

---

> ### Author Response · Authors · 2024-08-13
> **Further clarification and discussion.**
>
> Thank you very much for your timely and constructive comments. In response to your questions, we make the following clarifications:
>
> **Q1:** We apologize for any confusion caused by our previous responses. It appears the questions arose from our manuscript not clearly distinguishing our approach from existing methods. Before addressing our use of the term 'world model,' we'd like to clarify the key differences that set our method apart.
>
> Previous work (e.g., RT-2, RT-H, RT-X and RoboFlamingo) focused on using Visual Language Model (VLM) to facilitate language-guided robot manipulation tasks, improving the model's capabilities in scene perception, task planning, and logical reasoning. However, due to its lack of ability to model critical waypoints and asynchronous execution, the model learning ability and efficiency are low. Our PIVOT-R system is designed for complex language-instructed robot manipulation through a hierarchical approach. It begins by using a VLM to convert intricate commands and visual inputs into object-oriented skills (primitive actions). These actions guide the waypoint predictor (i.e., scene prediction module), which forecasts future waypoints for upcoming observations. The action prediction component then determines the necessary actions to reach these waypoints. By incorporating waypoints, PIVOT-R disentangles the dependency between language and actions, and better leverage cross-trajectory waypoint transition knowledge to improve the action prediction. Additionally, our model integrates an asynchronous hierarchical executor, making it suitable for real-time robotic control.  PIVOT-R offers a comprehensive framework that significantly enhances both effectiveness and efficiency.
>
> Now, we would like to explain the choice of the term "world model", specifically the “waypoint-aware world model”. As we've outlined above, the essence of our PIVOT-R system lies in its ability to predict waypoints, guided by the context of primitive actions. We initially believed that labeling it as a “world model” would provide an immediate and intuitive understanding for our audience, clearly illustrating the model's predictive capabilities based on actions and observations. However, we acknowledge the reviewer's feedback that using the term “world model” might inadvertently understate the complexity and depth of our contributions. To better highlight the uniqueness of our approach, we will reduce the description in terms of “world model” and emphasize PIVOT-R’s contribution to waypoint-aware modeling. We will revise our manuscript to reflect this change.
>
> **Q2: Fairness and cost of annotation.**
>
> There may be some misunderstandings here. Waypoint information in PIVOT-R is automatically generated and does not require any manual intervention. Specifically, we use the open source LLaVA 1.5 as the VLM for waypoint judgment, and use scripts to automatically generate waypoints in the robot's manipulation trajectory.
>
> **(1) Fairness.**
>
> We believe that fairness here mainly involves two aspects: experimental setting and dataset input. First, in terms of experimental settings, PIVOT-R and other methods maintain the same experimental settings such as optimizer and data augmentation. Secondly, in terms of dataset input, they all have the same input, and waypoint information is automatically generated by the VLM in PIVOT-R after receiving input. Since other baseline methods ignore the modeling of waypoint information, resulting waypoint information is useless for them. Therefore, we think their comparison is fair.
>
> **(2) Cost.**
>
> For waypoints information process, we used the open source LLaVA 1.5 as VLM on 8 RTX4090s to complete the waypoint generation of 13K trajectory data in 6 hours, which is affordable for practical implementation. Except for the manual definition of prompts (in Appendix F), no additional manual intervention is required. The prompts are simple and reusable.Therefore the cost is low.
>
> **Q3:** Thank you for your feedback, but we respectfully disagree with your method of evaluating the model's generalization ability. The key to evaluating generalization lies in how well a model performs under unseen conditions compared to others. PIVOT-R outperforms Surfer by 12.5%, 15.8%, and 15% in three different unseen scenarios. Evaluating models based solely on the percentage of performance drop can be misleading, especially when the initial performance levels are different. For example, if two students take a math exam, with Student A scoring 95 and Student B scoring 60, and then on a harder test, Student A drops to 85 (a 10.5% drop) while Student B drops to 59 (a 1.7% drop), it is clear that the absolute score is a more meaningful metric than the percentage drop.

---

> > ### Comment · Reviewer_9FHi · 2024-08-13
> >
> > Thank the author for the detailed response to my question and for pointing out some of my misunderstandings. After considering the opinions of other reviewers, I believe the author has addressed the reviewers' questions very well during the response phase, and I hope that the final version can integrate the new experiments and discussions into the paper. Therefore, I will raise my score.

---

> > > ### Author Response · Authors · 2024-08-13
> > >
> > > Thank you again for your constructive comments and help in improving our paper.

---

### Official Review · Reviewer_F1Qx · 2024-07-12

**Soundness:** 3
**Presentation:** 4
**Contribution:** 3
**Rating:** 7
**Confidence:** 4

**Summary:**

In this paper, the authors present PIVOT-R, a primitive-driven waypoint-aware world model for robotic manipulation. PIVOT-R comprises two key components: a waypoint-aware world model (WAWM) that parses primitive actions and predicts primitive-driven waypoints, and an Action Prediction Module that decodes low-level actions. Additionally, they introduce an Asynchronous Hierarchical Executor (AHE) for PIVOT-R, which enhances execution efficiency by applying different execution frequencies to different modules. Experimental results demonstrate that PIVOT-R achieves state-of-the-art performance on the SeaWave benchmark.

**Strengths:**

* The design of setting different execution frequencies for different modules (AHE) makes sense and enhances execution efficiency.
* The authors conduct extensive experiments, including comparisons with several baselines and ablated versions, and provide detailed discussion and analysis.
* The paper is well-organized and easy to understand.

**Weaknesses:**

- The proposed framework has only been evaluated in simulated environments. The authors are encouraged to conduct real-world experiments to demonstrate the method's sim-to-real transfer capability.
- The method relies solely on 2D RGB images without any depth information, which may hinder the accurate prediction of 3D robot actions, especially in unseen perspectives and scenarios. Additionally, transferring simulated RGB images to real-world RGB images without depth information is a non-trivial challenge.

**Questions:**

* I understand that different modules are assigned different execution frequencies, but what is the definition of the execution frequency? For example, when the authors set v1 = 3, v2 = 10, v3 = 30, does it mean that the VLM model is used 3 times and the action prediction module is used 30 times for each data sample? In addition, I am interested in knowing the average execution time for each module.
* The authors claim that world modeling of waypoints can prevent critical robot dynamics from being submerged in trivial robot manipulations. It would be beneficial if the authors could provide an ablation study or relevant examples illustrating this phenomenon.
* Given that the method operates in a closed-loop system, I wonder if there are cases where the robot can retry and successfully perform an action after an initial incorrect attempt? Or does an incorrect action always lead to task failure?
* Typo: In line 181, there is a repeated 'waypoint'.

**Limitations:**

The authors have discussed the limitations, but they are encouraged to provide and analyze specific failure cases of the proposed method.

---

> ### Author Rebuttal · Authors · 2024-08-07
>
> We are grateful for your comprehensive and encouraging review! We respond to all the issues you pointed out in detail below. We hope our response and rebuttal revision will address your concerns.
>
> **Q1. Real-world experiment.**
>
> We have added additional real-world experiments in Table 3 of the rebuttal PDF, where we set up three tasks: (i) "Pick up": pick up the correct object from the table. (ii) "Put on": Pick up the object and place it on the correct color block. (iii) "Push to": Push the object to the correct color block. For these three tasks, we collected 400, 200, and 200 sets of data respectively, for a total of 800 demonstrations. Finally, we perform 24 sets of tests for each task to calculate the average success rate. As shown in the results, compared with the best baseline Surfer, PIVOT-R achieved a **6% performance improvement**. With the help of the waypoint-aware world model, PIVOT-R has demonstrated excellent real-machine capabilities. We also added demonstrations of real-world evaluation in Figure 2 of the rebuttal PDF.
>
> **Q2. Action prediction of 3D robots.**
>
> The most advanced methods such as GR-1, RT-1, and Surfer are based on RGB, and for fairness and ease of comparison, we have followed their settings. Due to time constraints, we plan to introduce 3D information in our upcoming work.
>
> **Q3. Definition of execution frequency and execution time.**
>
> PIVOT-R's AHE executes the action prediction module once in each time step, and other modules are updated proportionally. That is, when the action prediction module is executed 30 times, the VLM and scene prediction modules are executed 3 and 10 times respectively. In addition, each inference time of the three modules of primitive action parsing, scene prediction and action prediction is about 177ms, 29ms and 18ms. The execution speed mainly depends on the action prediction module.
>
> **Q4. Too frequent waypoints cause the robot dynamics to be overwhelmed.**
>
> The corresponding experimental results are shown in row 3 of Table 3 (*i.e.*, PIVOT-R w/ next frame) in out paper. As shown in Table 3, compared with PIVOT-R (using key frames as waypoints), the average performance of PIVOT-R w/ next frame (using the next frame of the robot's manipulation trajectory as waypoints) dropped by 29.7% (74.19% vs. 44.45%). This shows that the waypoint-aware world model can effectively improve PIVOT-R’s modeling capabilities of key robot dynamics.
>
> **Q5. The result of a robot encountering an error during initial execution.**
>
> In some cases, retries may be successful. As shown in Figure 1 (left) in the rebuttal PDF, when the position of the gripper deviated and the object failed to be grasped, the second attempt to grasp was successful. However, in the case of Figure 1 (right), if the object is knocked down and rolls a certain distance, it will be difficult to successfully grasp it again.
>
> **Q6. Expression error.**
>
> Thanks for your reminder, we will modify this typo in our revised version.

---

### Official Review · Reviewer_eMP4 · 2024-07-13

**Soundness:** 3
**Presentation:** 2
**Contribution:** 2
**Rating:** 5
**Confidence:** 3

**Summary:**

This paper introduces PIVOT-R, an approach for language-guided robotic manipulation. PIVOT-R consists of a Waypoint-aware World Model (WAWM) and a lightweight action prediction module, along with an Asynchronous Hierarchical Executor (AHE) to improve efficiency. The model achieves state-of-the-art performance on the SeaWave benchmark, demonstrating improvements in performance and efficiency compared to baselines.

**Strengths:**

**Performance:** The proposed PIVOT-R model demonstrates strong performance improvements over chosen baselines, achieving state-of-the-art results on the SeaWave benchmark.

**Thorough Evaluation on Chosen Benchmark:** The authors provide comprehensive results on the SeaWave benchmark against strong baselines, and perform thorough ablations for generalization and interpretability.

**Weaknesses:**

**Lack of Clarity:** The abstract and introduction do not clearly articulate the specific problem being addressed. While they mention the need for language-guided robotic manipulation and the limitations of previous approaches, the exact nature of the problem, such as motivating the need for waypoint prediction in the first place, is not succinctly defined. Similarly, I found some  explanations of the experiments hard to follow in Sections 4.5 and 4.6. This could be improved by decreasing the use of lingo specific to the project and spending more time motivating each experiment before diving into details.

**Novelty:** The idea of breaking down tasks into action primitives and using waypoints is not entirely novel. Similar concepts have been explored in prior work, such as:

CLIPort: https://arxiv.org/abs/2109.12098

PerAct: https://arxiv.org/abs/2209.05451

RAPS: https://arxiv.org/abs/2110.15360

Learning Synergies between Pushing and Grasping with Self-supervised Deep

Reinforcement Learning: https://arxiv.org/abs/1803.09956

AWE: https://arxiv.org/abs/2307.14326

ForceSight: https://arxiv.org/abs/2309.12312

Scaling up and Distilling Down: https://arxiv.org/abs/2307.14535

Text2Motion: https://arxiv.org/abs/2303.12153

The authors do not mention most of these works, and they do not go into sufficient detail in distinguishing themselves from prior work. The paper also seems to lack reference to neurosymbolic/TAMP/PDDL approaches, which are concepts relevant to the idea of action primitives.

**Asynchronous Hierarchical Executor:** The AHE, while improving efficiency, appears to be a simple scheduler that can be implemented with very simple logic. This does not constitute a significant technical contribution.

**Questions:**

**Evaluation on Other Benchmarks:** Have you considered evaluating PIVOT-R on other well-known benchmarks to better contextualize the performance improvements? I may consider changing my rating if results on a more well-known benchmark are presented.

**AHE Mechanism:** Could you elaborate more on the algorithm behind the AHE module?

**Novelty:** Can you make a brief statement on the novelty of your work, particularly with respect to the prior work mentioned in the weaknesses section?

**Limitations:**

The paper discusses the limitations of the work, including the potential inconsistency between high-level instructions and underlying actions.

---

> ### Author Rebuttal · Authors · 2024-08-07
>
> Thank you very much for your detailed and constructive comments. We are delighted to see your recognition of PIVOT-R's state-of-the-art performance on the SeaWave benchmark. We respond to all the issues you pointed out in detail below. We hope our response and rebuttal revisions will address your concerns and lead to reconsideration of our work.
>
> **Lack of Clarity:**
>
> **(a) Waypoint prediction motivation.**
>
> In fact, we have a detailed and concise statement about the motivation of waypoint prediction in lines 3-9 of the abstract and lines 41-42 of the introduction. The motivation for waypoint prediction is mainly to prevent the model from being drowned in trivial scene and action predictions, which is crucial for world modeling and manipulation skill learning.
>
> **(b) Experimental motivation in Sections 4.5 and 4.6.**
>
> Section 4.5 discusses the impact of waypoint selection, VLM, AHE, scene prediction supervision, and action prediction module design on PIVOT-R's performance. Section 4.6 analyzes why PIVOT-R succeeds, exploring its generalization to new tasks and potential for further improvement by incorporating other datasets. These experiments are crucial for understanding each module's design effect and PIVOT-R's advantages.
> Thanks for your valuable suggestions, we will revise the relevant expressions and reduce specific terminology for clarity.
>
> **Novelty. Q1. Primitive actions and waypoints.**
>
> Thank you very much for providing so many valuable related works. We have noticed previous work on waypoints and primitive actions, noting that they often used a limited number of actions or lacked world models for support. For instance, CLIPort [1], Transporter [2], GMRT [3], and VPG [4] are restricted to simple actions like pick/place/push, limiting their use in complex tasks. Some language-guided models [5,6,7] define a few primitive actions (≤5) and add prompts to aid decision-making. In contrast, PIVOT-R supports 10 primitive actions, including unique actions like "rotate/open/close," making it effective in complex tasks. Crucially, PIVOT-R is the first primitive-driven, waypoint-aware world model, using primitives to break down tasks and combining scene and action prediction modules to model physical world knowledge. This enables PIVOT-R to handle complex tasks based on user instructions. We will also add the reference to the neuro-symbolic and PPDL approaches in our revision. Thank you again for your suggestion.
>
> [1] SHRIDHAR M, MANUELLI L, FOX D. CLIPort: What and Where Pathways for Robotic Manipulation[J].
>
> [2] ZENG A, PETE F, TOMPSON J, et al. Transporter Networks: Rearranging the Visual World for Robotic Manipulation[J]. arXiv: Robotics,arXiv: Robotics, 2020.
>
> [3] STENGEL-ESKIN E, HUNDT A, HE Z, et al. Guiding Multi-Step Rearrangement Tasks with Natural Language Instructions[J].
>
> [4] ZENG A, SONG S, WELKER S, et al. Learning Synergies between Pushing and Grasping with Self-supervised Deep Reinforcement Learning[C/OL]//2018 IEEE/RSJ International Conference on Intelligent Robots and Systems (IROS), Madrid. 2018.
>
> [5] COLLINS JeremyA, HOUFF C, TAN Y, et al. ForceSight: Text-Guided Mobile Manipulation with Visual-Force Goals[J]. 2023.
>
> [6] HA H, FLORENCE P, SONG S. Scaling Up and Distilling Down: Language-Guided Robot Skill Acquisition[J]. 2023.
>
> [7] Lin K, Agia C, Migimatsu T, et al. Text2motion: From natural language instructions to feasible plans[J]. Autonomous Robots, 2023, 47(8): 1345-1365.
>
> **Novelty. Q2. AHE's technical contribution.**
>
> It should be emphasized that the technical contribution of AHE is not isolated. It should be considered together with the waypoint-aware world model (WAWM). RT-H uses VLM for robot actions and language association, requiring VLM at every manipulation step, which reduces efficiency. In contrast, PIVOT-R's VLM-based primitive-driven WAWM for scene and action prediction, combined with AHE for asynchronous execution, improves efficiency by 28 times (Table 3 in our paper). Though simple, AHE's integration with WAWM is highly effective. We will emphasize this in the revised version. Thank you for the suggestion.
>
> **Questions. Q1. Performance of PIVOT-R on other benchmarks.**
>
> We evaluated PIVOT-R on the latest SIMPLER [1] benchmark, a scalable, repeatable and reliable proxy for real-world evaluation. We use this to verify the scalability of PIVOT-R in the real world. As shown in Table 1 of the rebuttal PDF, PIVOT-R outperformed the best baseline by nearly 10%. Additionally, we conducted real-world experiments, where we set up three tasks: (i) "Pick up": pick up the correct object from the table. (ii) "Put on": Pick up the object and place it on the correct color block. (iii) "Push to": Push the object to the correct color block.  We collected 400, 200, and 200 sets of demonstrations respectively. We tested each task 24 times to calculate the average success rate. As shown in Table 3 of the rebuttal PDF, PIVOT-R achieved a 6% improvement over the best baseline, Surfer. The waypoint-aware world model enabled PIVOT-R to demonstrate excellent real-world capabilities. Real-world evaluation demonstrations are included in Figure 2 of the rebuttal PDF.
>
> [1] Li, Xuanlin, et al. "Evaluating Real-World Robot Manipulation Policies in Simulation." *arXiv preprint arXiv:2405.05941* (2024).
>
> **Questions. Q2. AHE mechanism.**
>
> Specifically, we use multithreading to process each module separately. Each thread runs at its own frequency, extracts the latest data from the corresponding buffer, and places the output results in the buffer. For example, the VLM gets data from the camera buffer and saves the output in the buffer after each update. Then, the scene and action prediction module updates at different frequencies and reads the latest data from the cache of the previous module.  Different modules will not be blocked by other parts.

---

> ### Author Response · Authors · 2024-08-12
> **Waiting for further discussion**
>
> Dear Reviewer,
>
> Thank you once again for your valuable feedback on our work. In our rebuttal, we have provided detailed responses to the weaknesses and questions you raised.  We have clarified the novelty and distinctiveness of our waypoints and AHE. Additionally, we have elaborated on the experimental motivation. We have also conducted additional experiments, including SIMPLER benchmark and real-world experiment to better contextualize the performance improvements.
>
> As the author-reviewer discussion period is drawing to a close, we would like to check if there are any remaining questions or concerns that we can address. Please feel free to reach out if you need further clarification on any aspect of our work. We are committed to ensuring that our responses fully address your concerns and contribute to the improvement of our paper. We are confident that we have addressed all of your concerns and hope you will reconsider our work.

---

> > ### Comment · Reviewer_eMP4 · 2024-08-12
> >
> > Thank you for your detailed response. I appreciate the effort to clarify the motivation for waypoint prediction and to address the technical contributions of your work, as well as the additional evaluations on the SIMPLER benchmark and real-world tasks.
> >
> > **Lack of Clarity:**
> >
> > Regarding the motivation for waypoint prediction, I appreciate your point that it is meant to prevent the model from being overwhelmed by trivial scene and action predictions. However, I still find the argument somewhat vague. A clearer statement could emphasize that waypoint prediction enhances performance by creating a more meaningful mapping between instructions and actions, simplifying the task by focusing on key decision points rather than every low-level action. This would be analogous to how language models benefit from predicting at the token level rather than the character level, avoiding redundancy and focusing on more informative predictions.
> >
> > **Experimental Motivation:**
> >
> > Thank you for clarifying the experimental motivations. Most of this seems clear after rereading the paper. However, I think 4.6.1 and Figure 4 are still unclear. In particular, while $F_{O_t}$ and $F_{M'\text{t}}$ are defined, I don't think the definition of $F_{M_t}$ is mentioned in the paper. I think the discussion of this study should be grounded in less opaque language, such as "The observation feature approaches the waypoint feature as the task progresses". This would help to motivate the experiment.
> >
> > **Novelty:**
> >
> > The authors have provided useful references to prior work and clarified how PIVOT-R differentiates itself, particularly with the combination of waypoints and a world model. However, I still believe that the paper should include a more comprehensive discussion of related work, especially those that incorporate waypoints and action primitives. While the combination of a waypoint-aware world model and a greater number of primitives may be novel, the paper would benefit from a more thorough comparison with prior approaches to better highlight its unique contributions.
> >
> > I acknowledge your point that the Asynchronous Hierarchical Executor should be considered in conjunction with the Waypoint-aware World Model rather than as an isolated contribution. However, I maintain that the AHE, in its current form, seems to be more of an implementation detail than a key contribution. While it improves efficiency, its design as a simple scheduler using multithreading is not, in my opinion, sufficiently novel to warrant being highlighted as a major innovation. I suggest downplaying the emphasis on AHE as a key contribution in the paper.
> >
> > **New Results:**
> >
> > The additional evaluations on the SIMPLER benchmark and real-world tasks are appreciated and add value to the paper. These results give the reader more faith in the utility of your method.
> >
> > Based on your rebuttal and the additional results provided, I am inclined to raise my score from a 4 to a 5, contingent on the following revisions being made:
> >
> > 1. A section should be added to the related work that thoroughly discusses prior approaches involving waypoints and primitive actions, to better contextualize your contributions.
> >
> > 2. The emphasis on the Asynchronous Hierarchical Executor as a key contribution should be down-weighted, as it appears to be more of an implementation detail rather than a novel technical innovation.
> >
> > 3. Section 4.6.1 and Figure 4 should be made more clear.
> >
> > If these adjustments are made, I believe the paper would be stronger and warrant a higher score.

---

> ### Author Response · Authors · 2024-08-13
>
> **Q1: Clarity.** Thank you for your precious suggestions to help highlight the spotlight of our methods. And during the rebuttal process, we have revised our description to be more distinguishable. As noted by the reviewer, by introducing waypoints as a data structural chunking mechanism, similar to tokenization in NLP, we segment dense and irregular robot trajectories into meaningful sections, reducing the prediction burden. This hierarchical approach decouples language-action interdependencies and leverages cross-trajectory waypoint transition knowledge, improving action prediction accuracy. We will revise our manuscript to elucidate this concept more clearly. Furthermore, we believe this analogy could inspire future exploration of advanced tokenization techniques like Byte-Pair Encoding (BPE) to enhance language-instructed robot control systems. We will incorporate this discussion into our revised manuscript to provide a more comprehensive understanding. Once again, we are grateful for your insightful suggestions, which have greatly contributed to the depth and clarity of our work.
>
> **Q2: Experimental description.** Thank you for your feedback. $F\_{O\_t}$, $F\_{M'\_t}$, and $F\_{M\_t}$ represent the features of $O\_t$, $M'\_t$, and $M\_t$ respectively. We will add definitions of relevant terms. And we will improve writing in Section 4.6.1 with more concise and formulaic descriptions. For example, "The observation feature approaches the waypoint feature as the task progresses" will be revised into "The $L\_2$ distance between $F\_{O\_t}$ and $F\_{M\_t}$ gradually decreases as the task progresses". We will revise our manuscript to reflect these changes.
>
> **Q3: Novelty.** Thank you for your recognition and for providing so many valuable related works in previous discussions. We have conducted a thorough discussion and comparison among the related works in the latest version. In addition to the related work mentioned in previous responses, we also conduct a more detailed investigation. For example, PerAct[1], RVT[2] use robot states as waypoints to skip trivial action predictions. SUSIE[3] and UniPi[4] predict sub-goals through video predictors, but there is an inconsistency between the predicted video and actions. In contrast, PIVOT-R strategically selects waypoints by primitive actions to model physical world dynamics, thereby extracting the correlation between actions and scenes to enhance the accuracy of action prediction.
>
> [1] Shridhar, Mohit, Lucas Manuelli, and Dieter Fox. "Perceiver-actor: A multi-task transformer for robotic manipulation." Conference on Robot Learning. PMLR, 2023.
>
> [2] Goyal, Ankit, et al. "Rvt: Robotic view transformer for 3d object manipulation." Conference on Robot Learning. PMLR, 2023.
>
> [3] Black, Kevin, et al. "Zero-shot robotic manipulation with pretrained image-editing diffusion models." arXiv preprint arXiv:2310.10639 (2023).
>
> [4] Du, Yilun, et al. "Learning universal policies via text-guided video generation." Advances in Neural Information Processing Systems 36 (2024).
>
> **Q4: Emphasis on Asynchronous Hierarchical Executor.** Thank you for your suggestion. We will reduce the emphasis on AHE and add more implementation details in the experiment setting. We will improve this in the revised version.

---

> > ### Comment · Reviewer_eMP4 · 2024-08-13
> >
> > Dear Authors,
> >
> > Thank you for your detailed response. My main concerns have now been addressed, and I will maintain my increased rating of 5.

---

### Official Review · Reviewer_XYkg · 2024-07-15

**Soundness:** 3
**Presentation:** 3
**Contribution:** 2
**Rating:** 6
**Confidence:** 3

**Summary:**

The paper proposes PIVOT-R, a waypoint-based world model for robot manipulation. Concretely, given a language instruction, PIVOT-R converts it to a textual intermediate goal using a VLM and then feeds that as input into a scene prediction model that generates the waypoint. This waypoint is then fed into the action prediction model that predicts the robot action. The proposed method is compared with several baselines on the SeaWave benchmark and shows significant improvement in performance compared to numerous baseline methods.


Edit: The authors have addressed most of my comments.

**Strengths:**

- The paper seems to outperform the baselines by a margin on manipulation tasks of increasing complexity due to long range tasks
- The paper is well written and easy to follow
- The paper makes important technical contribution on effectively combining large VLMs with scene prediction models.
- Modulo some additional experiments (see below), the paper makes a strong empirical case for the proposed method (PIVOT-R) compared to baseline methods.

**Weaknesses:**

- The method is complex, and uses heuristics for defining intermediate waypoints, which does not look scalable at the outset.
- For baselines, it looks like there is video generation pretraining (GR-1) and Scene decoding (Surfer), however an important baseline is missing: SUSIE [A]
- Labeling of waypoints which is done in a heuristic way (zero hand speed, gripper state change, final frame, etc). It would be good to understand which is the most critical of these to get the most performance from the model.
- It would help to quantify the the additional labeling cost, and how scalable it is for various tasks.
- If I understand correctly, the baseline method Surfer which is the closest to the proposed method, seems to have 2 differences – a) splitting the task into waypoints, b) feeding the predicted scene as input. Ablation of these two changes to see which one contributes how much would help in understanding the paper.

[A] Black, Kevin, et al. "Zero-shot robotic manipulation with pretrained image-editing diffusion models." arXiv preprint arXiv:2310.10639 (2023).

**Questions:**

- Could the authors describe the ablation "PIVOT-R w/ video decoder" and provide an intuition on why might it hurt performance.

**Limitations:**

The authors should address the limitations in more detail. Specifically, discussing the additional labeling requirement of waypoints and the generalizability and efficacy of the heuristics to come up with the waypoints would help.

---

> ### Author Rebuttal · Authors · 2024-08-07
>
> We are grateful for your comprehensive and encouraging review! We respond to all the issues you pointed out in detail below. We hope our response and rebuttal revision will address your concerns.
>
> **Q1. The complexity and scalability of method.**
>
> **(i) Model design & implementation.**
>
> In fact, there is no complex and cumbersome design in PIVOT-R. In terms of design, the core module of PIVOT-R is a waypoint-aware world model, which is mainly composed of an open source VLM and a scene prediction model for primitive parsing and scene modeling. In terms of implementation, AHE sets different frequencies for different modules and adopts multi-threaded asynchronous implementation, which greatly improves execution efficiency. Therefore, the introduction of WAWM will not significantly increase the complexity of the entire model, and PIVOT-R is simple in design and implementation. We will also release our code upon acceptance of the paper to facilitate future research.
>
> **(ii) Heuristic methods for defining waypoints.**
>
> The heuristic method is a general waypoint definition method that has been adopted in many previous robot manipulation tasks [1-4]. PIVOT-R can achieve automated annotation of waypoints in various robot manipulation benchmarks using open-source VLM and scripts. Therefore, our waypoint annotation method has good scalability.
>
> [1] HSIAO K, LOZANO-PEREZ T. Imitation Learning of Whole-Body Grasps[C/OL]//2006 IEEE/RSJ International Conference on Intelligent Robots and Systems, Beijing. 2006.
>
> [2] AKGUN B, CAKMAK M, JIANG K, et al. Keyframe-based Learning from Demonstration[J/OL]. International Journal of Social Robotics, 2012: 343-355.
>
> [3] SHRIDHAR M, MANUELLI L, FOX D. Perceiver-Actor: A Multi-Task Transformer for Robotic Manipulation[J]. 2022.
>
> [4] JAMES S, DAVISON AndrewJ. Q-attention: Enabling Efficient Learning for Vision-based Robotic Manipulation[J]. arXiv: Robotics,arXiv: Robotics, 2021.
>
> **Q2. PIVOT-R *vs.* SUSIE.**
>
> Thank you for your suggestions. We compared PIVOT-R and SUSIE on the SeaWave benchmark, and the results are shown in Table 2 of the rebuttal PDF. As shown in the results, compared with SUSIE, PIVOT-R achieved an **average performance improvement of 26.93%** (74.19% *vs.* 47.26%) on four levels of SeaWave tasks. We found that this may be caused by the poor quality of the target image generated by SUSIE severely limiting the model's action execution when given unseen instruction input. In addition, compared with SUSIE, PIVOT-R still achieves a 9.17% (88.06% vs. 78.89%) performance improvement on Level 1 tasks without complex instructions. We think this is because SUSIE's diffusion model and low-level policy are trained separately, causing prediction deviations in the images that affect the low-level policy. In contrast, PIVOT-R's WAWM and action prediction modules are trained together, thus effectively avoiding this problem. We will add these results to the revised version.
>
> **Q3. Key factors in waypoint definition.**
>
> Thank you for your suggestion. We mainly consider three markers: zero hand speed, gripper state change, and primitive action completion frame to define waypoints. Among them, both zero hand speed and gripper state changes belong to robot state changes, and the robot arm speed is zero when the gripper state changes, indicating a strong correlation between the two. And they can judge directly through the robot motion status port. Therefore, as defined by Line 240-241 in the article, the above three waypoint judgment conditions can be divided into two types: primitive action completion frame and robot state change frame (including gripper and arm). To illustrate the contribution of the above two waypoint selections methods, we added a set of ablation experiments shown in Table 4 of the rebuttal PDF. As shown in the results, the performance of PIVOT-R with only primitive action completion frames dropped by 5.1%, the performance of PIVOT-R with robot state change frames dropped by 30.54%. Therefore, action completion frames are the main contributing factor.
>
> **Q4. Annotation cost and scalability.**
>
> **(1) Cost.** For dataset annotation, we use the open source LLaVA 1.5 as the VLM for waypoint (*i.e.*, primitive action completion frame) judgment, and use scripts to automatically annotate waypoints in the robot's manipulation trajectory. As shown in Table 3 in our paper, PIVOT-R is not sensitive to the choice of VLM. We used the open source LLaVA 1.5 as VLM on 8 RTX4090s to complete the annotation of 13K trajectory data in 6 hours, which is affordable for practical implementation.
>
> **(2) Scalability.** The annotation conditions of waypoints depend on the state changes of the robot and the visual changes of the manipulation trajectory in the simulator, which are common and easily accessible in different benchmarks. Therefore, this waypoint annotation method has good scalability.
>
> **Q5. Ablation experiment for Surfer.**
>
> The biggest difference between PIVOT-R and Surfer is that PIVOT-R adopts a waypoint-aware world model strategy. The results of using the next frame in Surfer as a waypoint in PIVOT-R have been shown in row 3 of Table 3 in our paper, which resulted in a very significant performance degradation (*i.e.*, performance loss of 29.7%, 74.19% *vs.* 44.45%). Taking the predicted scene as input is not the key difference between PIVOT-R and Surfer. Surfer and PIVOT-R essentially regard the robot action as the key factor in observing the image state transition, but they use different prediction orders. Thank you for this valuable comment, we will explain this in our revision.
>
> **Q6. Explanation of ablation for PIVOT-R w/video decoder.**
>
> The pixel-level prediction settings of "PIVOT-R w/ video decoder" are inconsistent at the semantic level with the high-level primitive actions such as "close to" and "grasp" that need attention. This results in degraded model performance.

---

### Author Rebuttal · Authors · 2024-08-07

We thank all the reviewers for their time, insightful suggestions, and valuable comments. We are happy that they appreciated our paper

- makes **important technical contribution** on combining large VLMs with scene prediction models (Reviewer XYkg);
- is **well-written, well-organized, and easy to follow** (Reviewer XYkg, F1Qx, and 9FHi);
- the design of the asynchronous hierarchical executor (AHE)  **makes sense and enhances execution efficiency** (Reviewer F1Qx);
- demonstrates **strong performance improvements** on the SeaWave benchmark and provides **thorough ablations** for generalization and interpretability (Reviewer eMP4, XYkg, and F1Qx).

We have also conducted additional experiments and provided necessary clarifications in our rebuttal, which are summarized below:

- In response to Reviewer eMP4 and F1Qx, we have added additional real robot experiments  (Table 3) and experiments on other benchmarks  (Table 1) in the rebuttal PDF. The results show that our PIVOT-R **significantly outperforms comparison methods** in both **real robot experiments and other benchmarks.**
- In response to Reviewer eMP4, we have highlighted our motivation and technical contribution of introducing waypoint prediction and asynchronous hierarchical executor, which are **crucial for enabling the world model to capture key dynamics and enhance execution efficiency**. We also provided a detailed analysis of the difference between our PIVOT-R and existing methods regarding action primitives and waypoint prediction. Note that both Reviewer XYkg and F1Qx **acknowledged the technical contribution of our paper.**
- In response to Reviewer 9FHi, we have clarified that **both the VLM and the scene prediction model** form the world model, which fits its definition of perceiving the environment and predicting changes in environmental states. We also highlighted that our core novelty is that we **introduce action primitives and waypoint prediction for improving world modeling**, which we believe is a valuable and inspiring idea for the robotic community. It is arbitrary to judge our novelty in simply action primitive decomposing and waypoint prediction.

We provide detailed answers to the reviewers' questions individually. We hope our response and rebuttal revision will address the reviewers’ concerns.

---

### Decision · Program_Chairs · 2024-09-25

**Decision:**

Accept (poster)

**Comment:**

This paper introduces PIVOT-R, an approach for language-guided robotic manipulation. PIVOT-R consists of a Waypoint-aware World Model (WAWM) and a lightweight action prediction module, along with an Asynchronous Hierarchical Executor (AHE) to improve efficiency. The model achieves state-of-the-art performance on the SeaWave benchmark, demonstrating improvements in performance and efficiency compared to baselines. The paper has been well written and presented. Overall, a good paper.
Authors have addressed most of the concerns raised by the reviewers during rebuttal. Please include some of the minor revisions in the future edit of the paper.